# Content Moderation and the Formation of Online Communities: A Theoretical Framework

## ABSTRACT

We study the impact of content moderation policies in online communities. In our theoretical model, a platform chooses a content moderation policy and individuals choose whether or not to participate in the community according to the fraction of user content that aligns with their preferences. The effects of content moderation, at first blush, might seem obvious: it restricts speech on a platform. However, when user participation decisions are taken into account, its effects can be more subtle — and counter-intuitive. For example, our model can straightforwardly demonstrate how moderation policies may *increase* participation and *diversify* content available on the platform. In our analysis, we explore a rich set of interconnected phenomena related to content moderation in online communities. We first characterize the effectiveness of the natural class of moderation policies for creating and sustaining stable communities. Building on this, we explore how resource-limited or ideological platforms might set policies, how communities are affected by differing levels of personalization, and competition between platforms. Our model provides a vocabulary and mathematically tractable framework for analyzing platform decisions about content moderation.

## CCS CONCEPTS

• **Information systems** → **Social networks**; • **Applied computing** → *Economics*; • **Theory of computation** → *Social networks.*

## KEYWORDS

Content moderation, social media, online platforms

**ACM Reference Format:**
Anonymous Author(s). 2023. Content Moderation and the Formation of Online Communities: A Theoretical Framework . In *Proceedings of ACM Conference (WWW'24).* ACM, New York, NY, USA, 24 pages. https://doi.org/10.1145/nnnnnnn.nnnnnnn

## 1 INTRODUCTION

Content moderation has been thrust to the center of public discourse about online platforms and their impacts on society. Once an esoteric task undertaken by social media companies with little external attention, content moderation is now seen (depending on whom you ask) as a means to fight misinformation, protect vulnerable communities, or manipulate public opinion.

In most jurisdictions, governments require platforms to moderate certain kinds of content, like child sexual abuse material and copyright infringement. However, platforms have wide latitude to choose moderation policies beyond this. In the U.S., platforms are not (as of yet) required to protect individuals' freedom of expression: the First Amendment only restricts the ability of government to limit speech, not that of companies. In addition, Section 230 of the Communications Decency Act protects social media companies from being sued for their decisions to host or remove a wide range of content (aside from what is required by law). In other countries, regulations vary, but platforms in general have significant discretion over their moderation policies. Content moderation in some form is a nearly universal feature of social media platforms [3], and a fundamental force affecting life online.

Naively, the effects of content moderation on platform speech are simple: content that violates platform policies is removed. However, when participation decisions are factored in, the effects may be more complicated. On the one hand, users who derive most of their enjoyment from banned content may decide not to participate, or they may move to another platform. On the other hand, mainstream users might *increase* their participation on the platform if they see less content they dislike. Thus, moderation policies will not have a uniform impact on individual decisions to participate and can lead to subtle and delicate interactions between user participation decisions. Indeed, changing membership in online communities may be key to understanding the societal impacts of social media. For example, Waller and Anderson [18] found a large influx of right-wing users was responsible for increased polarization on Reddit after the 2016 US presidential election. Similarly, online communities saw a massive increase in participation before the January 6 insurrection [16]. One goal of this work is to provide a framework to reason about how a platform's choices for its content moderation policy affect these dynamics.

Platforms seem to acknowledge that choices of moderation policies dramatically affect their ability to attract and retain users and communities. Founders of Gab, Parler, and Truth Social have marketed their platforms as free speech advocates, with less restrictive moderation policies. Similarly, during his acquisition of Twitter, Elon Musk promised to reduce the scope of the platform's moderation policies. In many other cases, highly restrictive rules are used to appeal to users seeking specific kinds of content, including contexts where political partisanship or toxicity are not typical concerns. For example, the subreddit r/aww features a moderation policy banning "sad" content so that it may better serve users seeking "cute and cuddly" pictures and videos.

Competition between multiple platforms adds further complexity to the dynamics of content moderation. Heterogeneity in moderation policies between different platforms may create a *market for rules* [7, 13]: platforms set rules and users can choose the platform with rules that best reflect their preferences. If a platform restricts speech that a user wants to engage in, then the user can leave and

join a competing platform in which that speech is allowed. Likewise, a user who encounters speech they find offensive can leave for a platform on which such speech is prohibited. This competition has led to striking changes to online communities, like the high-profile formation of Gab, Parler and Truth Social. Indeed, empirical evidence suggests that users deplatformed from Twitter migrated to Gab and saw increased activity and toxicity [1], although the overall size of communities deplatformed from Reddit and YouTube seems to have decreased [6, 14]. Content moderation policies are thus a key tool for platforms to survive and thrive, users critically shape the effect of these policies through their preferences and behavior, and these dynamics have important societal implications.

**The present work: modeling content moderation in online communities.** We present and analyze a simple, tractable model in which platforms set policies in order to build and maintain communities. The interaction between user participation decisions and content moderation policies is central to our framework and analysis, allowing us to explain fundamental and counter-intuitive phenomena about platform moderation decisions. For example, we can explain the basic fact, supported by empirical evidence [2], that moderation — even though it may deliberately remove some users — may foster much larger communities as a result: there are cases in which a platform can sustain a large user base under a carefully-chosen moderation policy while, without moderation, almost no users would participate. Similarly, our model can explain how the *range* of content available on a moderated platform may be greater than one without it.

In our model, content is associated with points in an ambient metric space; each user produces content from a subset of this space, and they also have a subset of this space corresponding to the set of content that they are willing to consume. For simplicity, we will restrict our attention to the case in which the ambient space is a one-dimensional axis, each user speaks from a single *speech point* and each user derives positive utility from content within an interval on this axis. Users derive utility when they encounter speech that they like and disutility when they encounter speech that they dislike; a user will join a platform if they would derive nonnegative utility from it and will leave otherwise. Notice that decisions to join, stay or leave create externalities for other users; when a user joins or leaves, their choice can change the utilities of other users (either positively or negatively), and this can potentially lead to a sequential cascade of arrivals or departures.

We next consider the content moderation policies available to a platform. We focus on a natural class of policies that satisfy a pair of properties:

(1) Speech-based: A moderation policy depends only on users' speech, not on what they prefer to consume.
(2) Convex: If speech at points $x$ and $z$ are allowed, then speech at any $y \in [x, z]$ should also be allowed.

Moderation policies that satisfy these properties can be specified by intervals: the platform deems a (possibly infinite) interval of speech permissible, and all users with speech points outside that interval are removed from the platform. We will call such an interval a *moderation window*, or *window* for short. We define and discuss the basic properties of the model in Section 2.

Using this model, we derive a series of results exploring a platform's ability to curate communities through its choice of moderation policy. In Section 3, we compare the effectiveness of window-based moderation to a theoretical optimum, the size of the largest set of users such that all users in the set have nonnegative utility with respect to the other users in the set. We also show that best window-based moderation policy can be approximated in a scalable way using a small sample from the population. In Section 4, we introduce an analysis of how much of a population a platform may lose if it does not implement moderation. Additionally, we discuss how platforms would choose moderation policies if *platforms themselves* have preferences over user speech. In Section 5, we study the effects of personalization systems on a platform's choices of moderation policies. We show that increased personalization may counter-intuitively decrease the size of an online community by shifting the distribution of content in ways that drive mainstream users away, even for the best choice of moderation window for a given level of personalization. Thus, our results provide a justification for limited personalization for some contexts even disregarding the challenges of predicting what users want. Next, in Section 6, we introduce and discuss a model of how competition among platforms may affect their moderation policies. Finally, we make several concluding remarks and discuss extensions of our model in Section 7. Proofs of our results are deferred to Appendix G.

In accordance with recent empirical evidence on the societal importance of changing membership patterns on social media [1, 6, 16, 18], we focus on the interactions of content moderation and user participation decisions. Our theoretical framework is naturally suited to explaining these empirical phenomena. Our work also offers a mathematically tractable way to explore a rich set of interconnected and subtle ideas in political philosophy, sociology and legal scholarship related to norms and political expression in communities. Here we list a few of these ideas. The *paradox of tolerance* posits that a tolerant society must be intolerant of intolerance in order to survive [12]; our model offers a mathematical interpretation of this idea, where a platform may ban extreme viewpoints that would otherwise drive other users off the platform. *Unraveling*: in sociology, there is a long tradition of studying cascading effects of participation in public activities; if certain members of a group representing a particular viewpoint begin to withdraw from public discourse, then others in this group might withdraw as well because they perceive themselves to be in the minority. Such dynamics are sometimes described evocatively as a *spiral of silence* [4]. In our model this corresponds to cascades of users, where once users with a particular viewpoint start leaving the platform, others of a similar viewpoint may also leave. Lastly, the *counterspeech doctrine*, outlined by the U.S. Supreme Court, says that the remedy to harmful speech is corrective speech [17]. Our model provides a framework to understand the conditions under which counterspeech can be effective and when harmful speech might overpower attempts at counterspeech. We believe that one of the strengths of our model is that it naturally captures the nuances of each of these concepts and provides theoretical explanations for when and why they occur.

**Related work.** Recent work has explored the interaction between shifting user beliefs and content moderation policies. Mostagir and Siderius [11] analyzes how platforms should set policies to reduce

the risk of harmful offline consequences as a result of extremist on-line forums. In their model, social media users joining a community assimilate to a common opinion. Platforms can either ban extremist communities or make them harder to find and engage with. They find conditions under which a policy that reduces the risks of an extremist event in the short-run backfires in the long-run and in which a policy that increases risk in the short-run may actually reduce it in the long-run. In a different paper, Mostagir and Siderius compare the effectiveness of various content moderation policies on the spread of misinformation in a network model of agents under (simple) DeGroot and (sophisticated) Bayesian opinion dynamics [10]. They show that moderation policies that work well for one type of agent may have the reverse effect for the other.

By contrast, in our work, we consider users' choices *whether to participate* in a community in the first place rather than modeling changing user preferences. Thus, our analysis is complementary and focuses how moderation policies shape the size and structure of online communities rather than individual beliefs.

Other recent work has analyzed how revenue models of social media companies might affect their moderation policies. Liu et al. [8] compares advertising- and subscription-based revenue models and finds that, in the regime where it is profit-maximizing not to moderate, advertising-based platforms are expected to host less extreme content. Otherwise, advertising-based platforms are predicted to be more extreme. Madio and Quinn [9] further explores the incentives of advertising-based social media companies in the presence of what advertisers consider "safe" and "unsafe" content.

Ours is a generalization of Schelling's Bounded Neighborhood Model [15], which was originally used to explore how mild in-group preferences of individuals within communities can lead to large disparities in their demographic composition, a phenome-non called *tipping*. In the original model, individuals come from two groups and have preferences for members of their own group. Groups in Schelling's model can be represented in our model by placing individuals in two stacks, one for each group; individuals of a given group have intervals identical to others in the same group and disjoint from those of the other group. Thus, our results can describe tipping where, in addition to two homogeneous demo-graphic groups, there may be a much greater range of identities and individual preferences over membership in the community.

## 2 A MODEL OF CONTENT CONSUMPTION, CREATION AND MODERATION

Our model of users in online communities starts with an ambi-ent metric space of potential content. Each user derives utility or disutility from content depending on its location in the space, and produces content at some location, called their *speech point*.

**User content creation and consumption.** We focus on the sim-ple case in which the content space is the real line and a user derives utility from content within an interval and disutility outside the interval. Formally, we define a *population* to consist of a finite col-lection of potential users, numbered $1, \ldots, n$. Each user $i \in [n]$ will have a speech point $p_i$ that is inside the interval $[l_i, r_i]$, which denotes the range of content from which they derive positive utility. A natural axis to consider is the left-right political spectrum, but an axis generically represents any dimension along which users

produce and consume content. Disutility might come from a user's belief that content is problematic, or they may just find some con-tent annoying, uninteresting, incomprehensible, or distasteful. A small population is depicted in fig. 1.

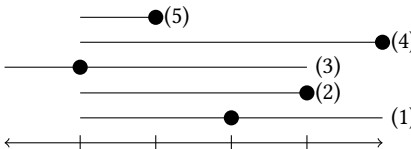

**Figure 1: Speech points and intervals of a small population.**

Our user model can succinctly capture significant user hetero-geneity: A user whose interval covers most other users may like to see a diversity of viewpoints on their feed; a narrow window might reflect a preference for consuming content similar to one's own belief. If the subject were, say, chess problems, a narrow win-dow might reflect a desire to wrestle with, and discuss, problems appropriate to one's level of skill. A user whose speech point is towards one end of their interval might like to challenge their own perspective from one direction or to keep tabs on an emergent or dangerous part of the content space; a user whose speech point is at the center of an interval might only care to consume content that is close enough to theirs, regardless of the direction of the disagreement. Many of our proofs and examples can reveal what user characteristics lead to particular aggregate phenomena.

**User utility and personalization.** For users $i, j \in [n]$, we will say user $j$'s content is *compatible* with user $i$ if $j$'s speech point lies in $i$'s interval: i.e., $p_j \in [l_i, r_i]$. We will say the users are *mutually compatible* if user $i$'s content is also compatible with $j$. Each user must choose to either use the platform or not use it. If user $i$ is on the platform, they will receive utility (normalized to) 1 from content they consume that is compatible with them and utility $-b_i$, where $b_i \in \mathbb{R}_+$, for content they consume that is incompatible with them. If they are off the platform, they receive zero utility.

In our model, platforms personalize content to each user by showing them content they like at higher rates than content they dislike. We call this *noisy personalization* since these systems do not necessarily show users only content they like: platforms may not have completely accurate classifiers, and some types of content, like replies to one's posts, may not be personalized at all by design. For a given user $i$, the platform will show them content inside their interval at rate (normalized to) 1 and content outside their interval with rate $\lambda_i \in [0, 1]$; i.e., the platform may filter out some of the content from which a given user derives negative utility. We can think of the user as consuming a sample of content where the relative probability they see a given piece of content inside versus outside outside their interval is determined by $\lambda_i$ and the relative proportions of content inside and outside their interval.

If the platform does not personalize at all, then the user sees content they dislike at rate $\lambda_i = 1$ and consumes a uniform random sample of the content on the platform. Unpersonalized forums (e.g., Facebook groups, subreddits or Discord servers) are particularly compelling motivating examples for our analysis, since they have been central to shaping impactful social phenomena, like the Jan-uary 6th, 2021 insurrection in the U.S. [16]. If the platform can

perfectly personalize to a user, then the user never sees content they dislike, and so $\lambda_i = 0$. This describes the context in which the user is in a perfect filter bubble and content moderation would not be necessary for creating and sustaining large, stable platforms; users would only see content from which they derive positive utility and so would always join. Noisy personalization is assumed to be an intrinsic, fixed quantity for each platform and user, determined by the platform's recommendation systems, fidelity of user signals, and other factors outside of the control of platform decision-makers choosing moderation policies.

Formally, if a set $\mathcal{S} \subseteq [n]$ of users is currently on the platform, user $i$'s utility from the content of $\mathcal{S}$ is calculated by

$$u_i(\mathcal{S}) = \sum_{j \in \mathcal{S} \setminus \{i\}} \mathbb{1}\{p_j \in [l_i, r_i]\} - \lambda_i b_i \mathbb{1}\{p_j \notin [l_i, r_i]\}.$$

Note that user $i$'s content is excluded from the content they can consume, so that a user's utility comes only from consuming the content of other users. If user $i$ is in $\mathcal{S}$, then $u_i(\mathcal{S})$ captures whether they choose to stay; if $i$ is not in $\mathcal{S}$, then $u_i(\mathcal{S})$ captures information about whether $i$ will join: if a user has nonnegative utility with respect to the set of users currently on the platform, they will join or stay on the platform; if they have negative utility, they will choose to leave or stay off the platform.

**Platform stability.** Let $\mathcal{S}$ denote the set of all users on the platform at a given instant. We say that this arrangement of $\mathcal{S}$ and its complement (those users not on the platform) is *stable* if all users $i \in \mathcal{S}$ have nonnegative utility $u_i(\mathcal{S}) \geq 0$ and all users $j \notin \mathcal{S}$ would have negative utility $u_j(\mathcal{S}) < 0$ if they joined the platform: a Nash equilibrium in a game where each player's strategy consists of the choice whether to participate and where each player's payoff is the utility they get as a result. Additionally, we will say that a set of users $\mathcal{S}$ are *compatible* if each user $i \in \mathcal{S}$ has nonnegative $u_i(\mathcal{S})$. (Unlike stability, compatibility does not say anything about the utilities of users excluded from $\mathcal{S}$.) In fact, we can express this model in a more succinct form: a user $i$ experiences nonnegative utility if and only if the fraction of speech on the platform (excluding user $i$) that is in their interval $[l_i, r_i]$ is at least $\theta_i := \lambda_i b_i / (1 + \lambda_i b_i)$, which we call their *participation threshold*. For the remainder of the paper, we will primarily use this equivalent, more succinct formulation.

How might a user's participation threshold $\theta_i$ vary across users and contexts? Platforms with better personalization for user $i$ (i.e., lower $\lambda_i$) will yield a lower participation threshold for user $i$, since $\theta_i$ decreases as $\lambda_i$ does. Users seeking out communities centered on organizing collective action might be expected to have a higher $\theta_i$: they might not participate if they see even a small fraction of dissenters and feel unsafe or unwilling to speak freely. Users seeking debate might be expected to have a lower value of $\theta_i$: they are willing to tolerate some content they dislike for the sake of healthy discussion. Thus, $\theta_i$ is an inherently content- and context-dependent parameter.

Throughout, we use the *platform* to refer interchangably to any single instance of a Facebook group, subreddit, TikTok's For You page etc. and *community* to refer to the (possibly changing) group of users on a platform over time.

**Moderation policies.** The primary focus of our analysis is on how content moderation policies affect online communities. We consider a simple class of content moderation policies that we will call *window-based moderation*, where an instance of the policy is a window. The window is a range of acceptable speech defined by the platform and will be represented by a closed interval: users whose speech is outside the interval are banned from the platform and those inside are given the choice to join or leave. A closely related concept from political science is known as the Overton window and defined by the range of positions that the public finds acceptable; the feasibility of a public policy depends on whether or not it is within this range of acceptable ideas. A window will be specified by $I := [l, r]$ or, equivalently, the set of individuals in the population whose speech points fall inside the interval. The set of window-based policies will be denoted $\mathcal{I}$.

**Initial adopters.** We assume the platform will start with some (possibly empty) set of *initial adopters* $\mathcal{S}_0 \subseteq [n]$, or the set of users who are already on the platform when the platform is choosing its moderation policy. Our analysis is applicable to platforms choosing moderation policies before any user has joined, as well as mainstream platforms which already have large user bases and are choosing a new moderation policy.

**Sequence of events.** A platform receives as input a population $P := \{l_i, p_i, r_i, \theta_i\}_{i=1}^n, \mathcal{S}_0$ (i.e., user intervals, speech points and participation thresholds and the set of initial adopters). The set of all populations will be denoted $\mathcal{P}$. The platform will choose a moderation window depending on $P$, and then users will be given the opportunity join or leave the platform one at a time in some infinite sequence, which we call their *switching order*. Note that the size of the community on the platform may depend on the switching order. For example, if user $i$ has negative utility on the platform, they will leave the platform given the chance. But if other users come before $i$ in the switching order and choose to leave first, then, by the time that $i$ gets the chance to leave, the composition of the platform has changed, and it's possible that they are no longer unhappy. We will denote a switching order $\sigma : \mathbb{Z}_{>0} \to [n]$, where $\sigma(t)$ is the user making the decision whether to switch at time $t$. Additionally, we will make the natural assumption that all switching orders are *starvation-free*; i.e., that at any time step in the process and for any user, there is a finite amount of time steps before they next decide to join, stay or leave.

**Extensions.** Natural extensions of our versatile model capture an even greater range of social media contexts. We derive many results for such cases in the appendix and discuss the extensions briefly in Section 7. We believe our model and results, with their many possible extensions, can provide a framework and vocabulary for exploring many different questions and contexts on social media.

## 3 HOW EFFECTIVE CAN MODERATION WINDOWS BE?

We begin by considering a single platform choosing a moderation policy and analyzing how large of a community window-based moderation is capable of achieving. To characterize the efficacy of window-based moderation, we compare against a natural baseline: the largest number of users that could all use the platform and derive nonnegative utility, a *largest compatible community* (LCC), i.e., a (not necessarily unique) largest set $\mathcal{S}$ such that $u_i(\mathcal{S}) \geq 0$

for all $i \in S$. Notice that an LCC is not necessarily stable: some users excluded from the LCC might derive nonnegative utility from participating on a platform with the LCC, but those users drive some of the utilities of LCC members below zero.

In general, the set of users at equilibrium for one switching order may be different from that of others. Indeed, there are examples of populations where the community never reaches an equilibrium for any starvation-free switching order. (These two facts are not unique to our context: for example, many games do not have a unique Nash equilibrium nor any pure-strategy Nash equilibria.) We provide and discuss results on the possibility of zero or multiple stable arrangements of users in Appendix F.

For a given population $P$ under a moderation policy $\pi$ and with switching order $\sigma$, the set of users at a time $t \in \mathbb{Z}_{>0}$ on the platform will be denoted $\mathcal{W}(P, \pi, \sigma, t)$. In our main results, we avoid making assumptions about particular switching orders. Instead, our results hold for all switching orders: we analyze a lower bound for a platform's size over any starvation-free switching order, defined as

$$s(P, \pi) := \min_{\sigma} \liminf_{t \to \infty} \left| \mathcal{W}(P, \pi, \sigma, t) \right|.$$

That is, for a given population $P$ and moderation policy $\pi$, $s(P, \pi)$ is the limiting minimum size of the platform over switching orders. Note that this definition is not sensitive to anomalies in the size of the platform that only occur finitely many times and that $s(P, \pi)$ is well-defined even if the platform never reaches an equilibrium. Thus, analysis of $s(P, \pi)$ does not require restrictive assumptions about the ways that user decisions interleave. Our guarantees hold no matter the order in which users make decisions.

Also, note that this is a true upper bound; i.e., $s(P, \pi) \leq s_{opt}(P)$ for any moderation policy $\pi$. To see this, note that, for any set of users $\mathcal{W}$ such that $|\mathcal{W}| > s_{opt}(P)$, there is a switching order in which users leave $\mathcal{W}$ until there are at most $s_{opt}(P)$ users remaining. Our goal in the remainder of this section will be to characterize the gap between $s(P, \pi)$ and $s_{opt}(P)$ when $\pi$ is a moderation window.

## 3.1 Window-based moderation vs. the largest compatible community: a special case.

To build intuition for window-based moderation, we begin with a special case before providing a general characterization of window-based moderation compared to the largest compatible community. Recall that the participation threshold $\theta_i$ governs what proportion of content user $i$ must see inside their window to join the platform. In this subsection, we consider the special case where $\theta_i = 1$ for all $i$, i.e., no user is willing to participate in a platform where any amount of content they consume falls outside their interval. This describes contexts where individuals are seeking out a like-minded bubble or where users could face serious repercussions if their membership in the group were known to outsiders, like those where members hold extreme views (e.g., if they were coordinating an insurrection) or are members of a targeted, vulnerable group (e.g., if they are seeking an abortion in a state where it is illegal).

In this special case, we show that moderation windows can fully recover the LCC and such a window can be found in polynomial time in the size of the population.

**Theorem 3.1.** *For any population $P$ where $\theta_1 = \cdots = \theta_n = 1$, there exists a moderation window $I^* \in \mathcal{I}$ such that*

$$s(P, I^*) = s_{opt}(P).$$

*Moreover, such a window $I^*$ can be identified in $O(n^3)$ time.*

We can explore this result through the example we introduced in fig. 1 (reproduced in fig. 2). Recall that user $i$ is compatible with user $j$ if $p_j \in [l_i, r_i]$; i.e., $i$ is happy to consume $j$'s content. In the example, the LCC is achieved with the set of users $\{1, 2, 3\}$. User 4 can't be included in the community because they are not compatible with user 2 or 3, even though user 4 would be content to engage on any platform with any subset of these users. User 5 can't be included because users 1 and 2 are not compatible with them.

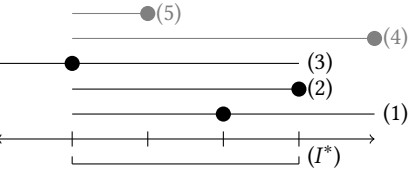

**Figure 2: A reproduction of fig. 1 (with LCC shaded in black). The window $I^*$ recovers the LCC.**

Notice that the speech points of all of users 1, 2 and 3 fall inside the window $[1, 4]$ and that their user intervals entirely cover the window. Thus, for any switching order and set of initial adopters, the platform can set the window to the interval $[1, 4]$ and eventually, only $\{1, 2, 3\}$ will remain. To see this, notice that when it is the turn of any of users 1, 2 or 3, they will join or stay on the platform, since they derive positive utility from any speech in the window. The main ideas behind proving Theorem 3.1 are similar to those we used to identify the LCC in our example. Every set of users where all are mutually compatible must have a particular structure: all of their intervals cover the entire interval between the left-most speech point and the right-most one. Thus, to find a window that can achieve a usership equal to the LCC, it is sufficient to find the most extreme speech points in the LCC and set the window to these extremes; any user in the LCC will have an interval that covers these extremes and a speech point inside the window. Any user in the LCC will be content to consume the content of any user in the window and any user in the window who does not cover the whole window will not be willing to consume the content of one of the users with the most extreme speech in the window and will leave on their own. There are only $O(n^2)$ different possible windows with endpoints equal to user speech points, and it only takes $O(n)$ time per window to check the number of users with speech point inside the window and interval that covers the window. Thus, the whole process only takes $O(n^3)$ time. The full proof is in Appendix G.1.

## 3.2 Window-based moderation vs. the largest compatible community: the general case.

In the $\theta_i = 1$ for all $i$ case, moderation windows suffice for the platform to capture the LCC, the theoretical upper bound on what any platform could achieve for a given population. One might hope that this holds more generally; however it turns out that when

users are more willing to consume some amount of content they disagree with (i.e., when $\theta_{\min} := \min_i \theta_i < 1$), the size of the community achievable with a window-based policy diminishes, relative to the size of the LCC. The first part of our next result generalizes Theorem 3.1 and says that moderation windows can capture a constant fraction of the size of the LCC for any population where $\theta_i$ is sufficiently large for all $i$. The second part says that this bound is tight, up to an additive constant: for some problem instances, the largest community achievable with window-based moderation is equal to our lower bound up plus a small additive constant.

**Theorem 3.2.** *For any population $P$ where $\theta_{\min} > 1/2$, there exists a moderation window $I \in \mathcal{I}$ such that*

$$s(P, I) \geq (2\theta_{\min} - 1)s_{opt}(P)$$

*that can be identified in $O(n^3)$ time. On the other hand, for all $\theta_{\min} > 1/2$, there exists a population $P$ such that for any moderation window $I \in \mathcal{I}$, it holds*

$$s(P, I) \leq (2\theta_{\min} - 1)s_{opt}(P) + O(1).$$

*and, for all $\theta_{\min} \leq 1/2$, there exists a population $P$ such that for any moderation window $I \in \mathcal{I}$, it holds $s(P, I) = O(1)$.*

In other words, first, the size of a community achievable with a window-based policy is no less than a $(2\theta_{\min} - 1)$ fraction of the largest compatible community. It also says that there exist populations where a window-based policy can achieve at most a $(2\theta_{\min} - 1)$ fraction of the largest compatible community (up to a small additive constant), and if $\theta_{\min} \leq 1/2$, a window-based policy is only capable of achieving a constant number of users. This is a tight lower bound on how effective a platform can be in terms of the participation thresholds of users, relative to the LCC.

Theorem 3.2 tells us that when users have lower participation thresholds, meaning they are relatively more willing to see a greater proportion of content that brings them disutility, it may be counter-intuitively harder to set policies that preserve a large fraction of the large community. On the other hand, when users have higher participation thresholds, platforms can choose simple policies that preserve a large fraction of the largest compatible community. In-tuitively, this result holds because every population for sufficiently large $\theta_i$ for all $i$ contains an efficiently-identifiable set of mutually compatible users using the same algorithm used to prove Theorem 3.1. In real world platforms, this corresponds to anchoring a large, core set of users on the platform by ensuring they will like everything they see, while potentially banning other users who would like to participate in the platform but might not be compatible with users in the core set. Next, we consider how moderation windows can be identified on very large populations.

### 3.3 Is it possible to find approximately optimal moderatation windows on very large platforms using limited resources?

In Theorem 3.2, we showed that with moderation windows, a platform can capture a constant fraction of the largest compatible community when $\theta_i$ is sufficiently large for each user $i$. However, the algorithm to choose such a window requires $O(n^3)$ time, which may be infeasible for large platforms. Can a large platform efficiently

set a moderation window to capture a large user base? We show in Theorem 3.3 that the answer to this question is yes: with just a small sample of users, a platform can construct a moderation window that is within a constant factor of $s_{opt}(P)$ with high probability.

**Theorem 3.3.** *For any population $P$ such that $\theta_i > 1/2$ for all $i$, it holds that, for any $\beta \in (0, 1)$, given a random sample of $m$ users, a platform can find a moderation window $I \in \mathcal{I}$ in time polynomial in $m$ such that*

$$\Pr\left[s(P, I) \geq \beta \cdot (2\theta_{\min} - 1)s_{opt}(P)\right]$$

$$\geq 1 - n \exp\left\{-m\left(\left(\theta_{\min} - \frac{1}{2}\right)\frac{s_{opt}(P)}{n}(1 - \beta)\right)^2\right\},$$

*where the probability is over the sample of $m$ users. In fact, sampling $m$ users uniformly at random and applying the window achieving the lower bound in Theorem 3.2 on the sample to the full population will achieve the stated bound.*

In other words, it is possible to capture a constant fraction of the guarantee for window-based moderation in Theorem 3.2 with high probability using a sample of $m$ users. In fact, it suffices to have $m = O(\log n)$ samples to get a non-trivial guarantee as long as $s_{opt}(P) = \Omega(n)$ (i.e., there exists a compatible community consisting of a constant fraction of the population). Intuitively, this is possible because a moderation window that achieves a large community on a random sample of the population is likely to also achieve a large community on the entire population. In fact, the largest community achievable with a window in the sample is likely to "cover" the largest community achievable with a window in the population in the sense that the most extreme speech in the sample are distributed cloe to the most extreme users in the largest community achievable with a window in the population.

The theorem shows us that, as the platform scales, moderators need not necessarily do anything complicated to learn effective policies: they can simply sample from the users, compute a policy with respect to the sample, and apply it to the whole platform while preserving a constant fraction of the size of the platform, with high probability. We defer the proof to Appendix G.1. We also note that our results throughout Section 3 do not depend on the set of initial adopters $\mathcal{S}_0$: our results apply to situations where the platform starts empty, full or anywhere in between.

## 4 THE DECISION TO MODERATE

In Section 3, we explored the degree to which a natural class of platform moderation policies might be used to curate a large community on a platform, relative to the theoretical optimum for the population, and how they might do so just from a random sample of the population. Now we consider platforms that can't or won't impose moderation policies on their users, and platforms that have ideological preferences over user speech.

To illustrate, suppose a platform is unwilling to pay the costs of setting up and maintaining a moderation system or the platform is so-called *free speech absolutist*, imposing a normative belief that there should be no platform-imposed boundaries on acceptable speech. If the platform imposes no moderation policy, the size of the community it can support might be dramatically smaller than if it moderated: some individuals might be exposed to too much speech

they find objectionable and leave. It is simple to find populations where this choice is disastrous: the platform cannot maintain a stable community of more than a constant number of users, even if, with moderation, it could maintain a user base of size $\Omega(n)$. All it takes is a constant number of users (in the size of the population $n$) who nearly all others find disagreeable in order to prevent the platform from attracting and sustaining a large community. This is formalized in our next result.

**Proposition 4.1.** *Consider any population $P$ where there exists a nonzero constant number of users whose speech point falls outside the intervals of all other users (i.e., $i \in [n] : p_i \notin [l_j, r_j]$ for all $j \in [n]$). Then for all moderation windows $I \in \mathcal{I}$ such that $p_i \in I$ and empty set of initial adopters $\mathcal{S}_0 = \varnothing$, it holds that $s(P, I) = O(1)$.*

In other words, if there a small number of users, *trolls*, who all others find intolerable, and the troll joins the platform first (as they will for some switching orders), then no other users will join. On the other hand, it is trivial to find populations where, setting a window that excludes the trolls, every other user would be content to participate on the platform, for any switching order. This provides a pragmatic argument for content moderation: even if a platform has no preferences over the speech it hosts, it may still engage in moderation for the sake of self-preservation. Without moderation the platform will be effectively abandoned. The same is true for the *range* of speech available with and without moderation. Without moderation, the range of speech available on the platform would be limited to the trolls' (which may just be $p_i$ if the troll is user $i$), while with moderation, a much larger range of speech could be available (from $\min_i p_i$ to $\max_i p_i$). We explore these results in detail in Appendix A.1.

More generally, a platform's owners may have *ideological* preferences for some ranges of speech over others. In some cases, advertisers do not want to be associated with certain "extreme" content, meaning a platform only earns ad revenue from some of its users. Thus, it is natural to imagine that the *platform's* utility may depend on the speech points of the users it hosts. As with users, we can imagine that the platform has some interval within which they derive positive utility (normalized to) 1 and outside of which they derive negative utility $d$. We might think that for a platform with such preferences, the optimal strategy would simply be to set its moderation window equal to its speech preference, banning users whose speech points fall outside of this interval. Intuitively, this makes sense: the platform would ban any speech from which it gets negative utility. However, as we explore in Appendix A.2, this strategy does not necessarily maximize the platform's utility. We demonstrate how it may instead be optimal for a platform to set a window that is *narrower* or *wider* than the interval within which they derive positive utility from user participation. Our results reveal limits to a platform's ability to act without consideration for the delicate relationships between user preferences; it must make some effort to cater to its users.

## 5 CONTENT MODERATION AND PERSONALIZATION

In the preceding sections, we explored how and why platforms might set moderation windows to create and maintain communities

online. In this section, we consider how changes to personalization systems affect platforms' ability to sustain large communities.

It is natural to ask how a population subject to different personalization systems may be different. In particular, it would be intuitive to expect that personalization is an unmitigated good from the perspective of the size of a community a platform could support: the same population consuming content that is more personalized should be more willing to participate in the platform than if they were consuming content that was less personalized. In a very limited sense, this is true. For any fixed set of users, the theoretical upper bound on the size of the community a platform can support (i.e., the largest compatible community) is nondecreasing as personalization improves. Thus, the *largest a platform could possibly be* only improves with better personalization. We formalize this result in Proposition 5.1

**Proposition 5.1.** *Consider two populations $P := \{l_i, p_i, r_i, b_i, \lambda_i\}, \mathcal{S}_0, P' := \{l_i, p_i, r_i, b_i, \lambda_i'\}, \mathcal{S}_0$ that differ only by the fact that personalization in the second is no worse for any user than in the first: i.e, $\lambda_i \geq \lambda_i'$ for all $i \in [n]$. Then it holds that*

$$s_{opt}(P) \leq s_{opt}(P').$$

The result tells us that the LCC is monotonically nondecreasing as personalization improves. This proposition follows from the fact that a compatible set in $P$ is also compatible in $P'$; if you have a set of compatible set of users and improve personalization, the set of users remains compatible.

The story gets more complicated for platforms trying to implement content moderation policies using window-based policies and accounting for switching dynamics; when we account for the delicate interactions between user preferences, a platform with better personalization may not necessarily be better off. Namely, we can construct a class of populations where increased personalization leads to a reduction in the size of the platform relative to the same set of users subject to less personalization, even if the platform chooses the best moderation window for a particular level of personalization. We formalize this result in Proposition 5.2.

**Proposition 5.2.** *There exist two populations $P := \{l_i, p_i, r_i, b_i, \lambda_i\}, \mathcal{S}_0, P' := \{l_i, p_i, r_i, b_i, \lambda_i'\}, \mathcal{S}_0$ that differ only by the fact that personalization in the second is strictly better for every user than in the first (i.e, $\lambda_i > \lambda_i'$ for all $i \in [n]$) such that*

$$s(P, I^*) > s(P', I^{*'}) \tag{1}$$

*for $I^*$, $I^{*'}$ optimal choices of window-based policies on $P$ and $P'$ respectively. In fact, such a pair of populations satisfying ineq. (1) can be constructed for any $b := b_1 = \cdots = b_n$ and $\lambda := \lambda_1 = \cdots = \lambda_n$ and an appropriately chosen $\lambda' := \lambda_1' = \cdots = \lambda_n'$.*

To illustrate how this might work in a real-world context, consider a forum dedicated to a professional sports league. Such a forum may permit the largest membership when most users are generally interested in a variety of discussions of highlights, news and events surrounding the league. Also suppose there is a small minority of fans of a particular team who are only interested in news and events surrounding their team. With limited personalization, the team-specific minority might be unwilling to participate if there is not a critical fraction of content about their team. Under more personalization, however, these team-specific fans may be drawn onto

the platform, leading to a disproportionate amount of team-specific content in the forum. This can trigger a cascade of general-interest off the platform, leaving only the smaller group of team-specific fans. This all occurs despite the fact that all user speech was within the bounds of the best window for each level of personalization. Proposition 5.2 provides a justification for why highly personalized recommendations are not always advantageous, even putting aside the difficulty of building personalization systems.

## 6 CONTENT MODERATION UNDER COMPETITION

In many real-world contexts, platforms face competition for users, and as others have noted, this competition might shape how they set moderation policies, leading to a *market for rules* where users participate in communities with the rules they prefer [7, 13]. Recent, high-profile examples have shown how alternative platforms try to challenge mainstream ones on the basis of their moderation policies: platforms like Gab, Parler, Truth Social, Bluesky, Mastodon and Threads each emerged to try to attracting users away from Twitter or capitalize on communities banned by Twitter's content moderation. We model this competition by considering multiple platforms who can each set moderation policies, and users choose which (if any) platform to use.

In Appendix B.1, we formalize a model of competition for users. Our formalization hinges on the scarcity of user attention: without scarcity, platforms would just be solving independent moderation problems. We term this scarcity users' *bandwidths*, define it by the amount of content on a platform a given user can consume.

Then, we explore how insurgent platforms, such as the new platforms above, may try to compete with an existing, incumbent platform, like Twitter, on the basis of their moderation policies. This may entail allowing users who are banned from the dominant platform and perhaps enticing other users to follow them to the new platform; alternately, it might involve setting *stricter* policies to appeal to users dissatisfied with the speech on the dominant platform. We characterize how, depending on the population, the incumbent platform may have wide latitude in choosing their moderation policies without regard to the risks of losing their membership to a competitor or may be highly vulnerable to competitors regardless of their policy. The analysis is in Appendix B.2.

We also extend Section 5 to explore differing levels of personalization on multiple platforms. With multiple platforms, these personalization systems may be heterogeneous, either because of inherent differences in the systems themselves, the amount of effort they have allocated to building recommendations, or some other reason. In Proposition B.1, we show how, even factoring in the dynamics of users switching between platforms, a platform with worse personalization can out-compete a platform with better personalization, even if the platform with better personalization starts off with an LCC on the platform. This result complements Proposition 5.2 because, rather than considering counterfactual platforms each setting windows in isolation (as in Proposition 5.2), we consider two platforms that simultaneously exist and for which users must choose to participate on one platform or abstain from both. The analysis is Appendix B.3.

## 7 CONCLUSION AND EXTENSIONS

In this work, we provide a framework in which to analyze content moderation, user participation decisions and their impacts on communities. Taking a natural class of moderation policies, moderation windows, we characterize how effectively these policies can create and sustain large communities. We also show how our model can provide insight about platforms' ideological preferences, differing levels of personalization, and competition between platforms.

Our model can be naturally extended to explore a number of related phenomena and contexts on social media. These extensions shed further light on various social media contexts that can be described by building on our basic model. Here, we describe several such generalizations, providing references to the appendix where we have derived relevant results.

**Population changes.** An important consideration for platforms deciding to implement policies is how robust they are to unexpected changes to the population. In Appendix C, we consider a variant of the model where the population experiences an $O(1)$-sized change after the moderation policy is set. We explore the *price of robustness*: how much smaller a platform would have to be to ensure a stable platform after an $O(1)$-sized change to the population.

**Dynamic moderation policies.** Moderation policies may change over time, and platforms might want to roll out a squence of policies to maximize their user base, even if the population does not change. Dynamic window-based moderation would allow for platforms to set a moderation window that changes over time. In Appendix D, we explore an interesting class of populations where dynamic windows are strictly more powerful than static windows.

**Lurkers and heterogeneous speech frequencies.** Not all members of a community need to create content in order to participate. Our model can be extended to encompass heterogeneity in speech frequencies by associating a frequency $f_i$ to each user. This generalization allows for analysis of an additional type of content moderation: caps on the speech frequency for each user. We consider this variant of the model in Appendix E and present a reduction from the problem of computing the size of (an appropriate definition of) the LCC for this generalization to the size of the LCC for a carefully constructed fixed-frequency problem instance.

**Other utility functions.** In our work, we consider the natural special case of constant positive utility from consuming content inside the user's interval and constant negative utility outside, i.e., users either "like" or "don't like" content from each other user but don't otherwise make distinctions between different content. Our user utility model could be naturally extended to other functions over the content space; indeed, several of our results could be generalized by redefining "compatible" as content from which a given user derives nonnegative utility. One could also consider user *production* utilities; users may derive utility from producing content that others like to consume.

**Multiple dimensions.** It would be useful to extend the ambient content space to multiple dimensions, rather than just one: e.g., economic versus social policy discussions might be best modeled by a two-dimensional ambient space. Other discussions might be best modeled by a high-dimensional latent embedding space.

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

# A  THE DECISION TO MODERATE

We provide a more complete analysis and discussion of Section 4.

## A.1  When is moderation necessary?

With no moderation, the platform has no control over its users; they simply join or leave according to some switching order. In Proposition 4.1, we state that all it takes is a small number of trolls in order to prevent the platform from attracting and sustaining a community of more than a constant number of users.

Let $i$ be the index of any user whose speech point falls outside the intervals of all other users. To see why the proposition holds, notice that user $i$ is first in the switching order. User $i$ will join the platform, since the first user on a platform will have zero utility, and then no other users will join. Indeed, this is true even for populations where all other users would join the platform for any switching order. For example, consider a population $P$ where, for some $l, p, r$, it holds that $l = l_1 = \cdots = l_{n-1}$, $p = p_1 = \cdots = p_{n-1}$ and $r = r_1 = \cdots = r_{n-1}$ and $p_n = r + 1$. That is, all user preferences are identical except for those of user $n$. If user $n$ joins the platform (as they will for some switching orders), none of the other users will join the platform. However, a moderation window were set to $[l, r]$, then all users $1, \ldots, n-1$ would join the platform. Thus, as $n$ grows, the gap between the size of the moderated platform and an unmoderated one can grow arbitrarily large.

In fact, the conditions for a large gap between a moderated and unmoderation platform stated in the proposition are not unique: for example, the platform does not necessarily need to start empty. Any number of combinations of number of trolls, sets of initial adopters and switching orders can be conceived where a small number of trolls force the rest of the users off the platform.

Proposition 4.1 provides some insight as to why platforms remove users even when their ultimate goal is to maximize their user base: without moderation, a platform may hollow out as some users drive others away. Moderation windows allow platforms to build stable communities.

## A.2  How might an ideological platform choose policies?

In this subsection, we will consider moderation policies of ideological platforms (i.e., platforms whose intervals are not $(-\infty, \infty)$). As with users, we will imagine that a platform has some interval $[l, r]$ within which they derive positive utility (normalized to) 1 and outside of which they derive negative utility $b$. When the platform's interval is $(-\infty, \infty)$, the platform derives utility from all speech; this corresponds to the objective of maximizing the size of the platform, which aligns with the focus of our analysis outside of this section.

We might think that for a platform with some interval $[l, r]$, the optimal strategy would simply be to set its moderation window equal to $[l, r]$, banning users whose speech points fall outside of this interval. However, this strategy does not necessarily maximize the platform's utility. In fact, we already saw an example of this in Proposition 4.1. Even though the platform derives positive utility from users participating anywhere on $(-\infty, \infty)$, it may be utility-maximizing set a narrower moderation window. In this example, the narrower window was necessary to ensure that more extreme users do not drive others off the platform, even though the platform itself had no preference against extreme speech points.

In other cases, it may instead be optimal for a platform to set a window that is *wider* than their interval. Suppose a platform consists of four types of users as in fig. 3: there are far-left, center-left, center-right and far-right users. We use the notation $\times 0.25n$ here and in future examples to indicate the number of users whose speech point is of that type. Users of type 1 derive positive utility from all other users. All other users derive utility from users of the same type and users directly to their left. The platform derives utility 1 from each user on the platform within the interval containing the center-left, center-right and far-right users and disutility $d$ for each user off the platform for some $d < 2$. Suppose that the set of initial adopters is empty: $S_0 = \varnothing$.

What would happen if the platform simply set its moderation window equal to its interval? If it did, any switching order in which

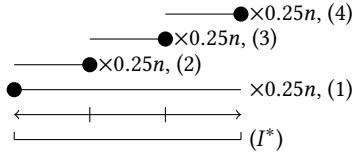

**Figure 3: An ideological platform that sets a window wider than its interval for $\theta = 1/4, d < 2$.**

users of type 4 start would end up with only users of type 4. Meanwhile, if the moderation window were set to include all users, then users of type 1 would always join the platform, since they are compatible with all others. Then, once all users of type 1 were on the platform, all users of type 2 would also join the platform, regardless of any other users were already on the platform. Similarly, all users of type 3 would then be willing to join, and finally users of type 4 would join. While they would suffer disutility $d \cdot 0.25n$ for the presence of type 1 users, this would be more than offset by the utility gain of $0.5n$ from type 2 and 3 users who are now willing to remain on the platform, since by assumption $d < 2$. It is advantageous for the platform to allow all users on the platform so that users of type (2) and (3) join the platform.

These examples demonstrate that even if a platform has preferences over speech, the optimal strategy to implement those preferences may depend on the user population. This can include setting a narrow window to prevent users from being driven off, setting a wider window to retain users who enjoy speech that the platform does not, or potentially some combination of the two.

# B CONTENT MODERATION UNDER COMPETITION

We provide a more complete analysis and discussion of Section 6.

## B.1 A model of consumption bandwidth under competition.

If platforms are going to compete for users, then user attention must be scarce: otherwise multiple platforms would just be solving independent moderation problems on the same population — and user $i$ could just participate in all platforms where they would derive positive utility. Thus, scarcity of user attention is a crucial element that we will add to the model to introduce competition between platforms. We will call this scarcity the users' *bandwidths*.

Formally, we assume that each user has some bandwidth $\gamma_i > 0$ for consuming content. The parameter $\gamma_i$ represents the number of users on a platform for whom user $i$ can consume all content. If a platform has greater than $\gamma_i$ users and user $i$ is using the platform, they will consume a random sample of the content, which we model deterministically for simplicity as the expected utility derived from consuming $\gamma_i$ users' content.

Recall that each platform has a personalization system that may filter out some of the content that a user derives disutility from. With multiple platforms, these personalization systems may be heterogeneous: an individual's personalization on one platform may be different, either because of inherent differences in the systems themselves, the amount of effort they have allocated to building

recommendations, or some other reason. Thus, user $i$ on platform $j$ will be said to have personalization parameter $\lambda_{i,j}$ on that platform.

If a user is choosing between two platforms from which they would derive positive utility and for which the size of both platforms is greater than $\gamma_i$, the user will go to the platform for which they are compatible with a higher fraction of content they consume. We call this *proportion-based switching* because users are choosing the platform for which they consume a higher proportion of content compatible with them. (Recall that some of the content outside their interval may be filtered out, so individuals may not consume a representative sample of the composition of the platform as a whole).

On the other hand, if one or both of the platforms are smaller than size $\gamma_i$, we assume users take into account total utility, rather than just the proportion of compatible content, when choosing a platform. We call this *utility-based switching*. Notice that proportion- and utility-based switching smoothly interpolate: For each platform, they calculate the proportion of content inside their interval minus the rate of content outside their interval $\lambda_i$ times the disutility from consuming content $b_i$ all times the minimum of their bandwidth and the size of the platform, and then go to the platform with larger utility. That is, if the current sets of users on the platforms are $S_1 \ldots S_k$, then the utility user $i$ would receive on platform $j$ is defined as

$$v_{i,j}(S_j) := \min\left\{\gamma_i, |S_j|\right\} \sum_{\ell \in S_j} \mathbb{1}\{p_\ell \in [l_i, r_i]\} - \lambda_{i,j} b_i \mathbb{1}\{p_\ell \in [l_i, r_i]\}$$

where we assume the individual chooses the platform that derives them higher total utility and they produce their content on this platform. As in the preceding sections, platforms are *stable* if no user stands to gain utility from changing from their current position while the other users stay the same. We will also specify as starvation-free switching order as before and a set of initial adopters for each platform.

As in the single platform case, platforms need not necessarily reach an equilibrium. This can be the case under either proportion- or utility-based switching. However, in some circumstances, such as when all compatibilities are mutual under utility-based switching (i.e., user $i$ is compatible with $j$'s content if and only if $j$ is compatible with $i$'s content) then it is possible to show that the platform will always reach an equilibrium. Details on cycling for multiple platforms are available on Appendix F.2.

## B.2 Emergent platforms challenging a large, stable platform.

We explore populations where a single platform can sustain a large stable set of users in isolation and a second platform is started to try to attract as many users as it can. To formalize this, we will assume platform 1 starts with a large stable set of users, and platform 2 starts empty. Through two examples, we show that, depending on the population, a large, previously stable platform can be extremely vulnerable to a new platform and that in other cases, the platform may have wide latitude to choose a moderation policy — even a sub-optimal one in the sense of the largest number of users it could attract — while still preventing the new platform from attracting

many users. The first situation we call *insurgency bias* and the second situation we call *incumbency bias*.

**Insurgency bias.** Large platforms choosing best windows can be vulnerable to disruption, even if the large platform has nearly all of the population on their platform. We can see this in fig. 4, fixing any $0 < \varepsilon < 1/2$. Suppose the set of initial adopters for the first platform is all users: $S_1 = [n]$.

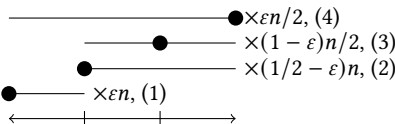

**Figure 4: A population where there is insurgency bias for** $\theta := \theta_1 = \cdots = \theta_n = 1/2, 0 < \varepsilon < 1/2$.

First, notice that the first platform can set a window so as to capture users of type 1, 2 and 3, which constitute $(1 - \varepsilon/2)n$, nearly the whole population if $\varepsilon$ is close to 0. If the platform does not set a window to exclude users of type 4, users of type 1 would have $(n/2 - 1)/(n - 1) < 1/2$ compatibilities, so they would exit the platform given the chance, leading to a smaller platform after it reached equilibrium. However, when an alternative platform becomes available, the alternative platform's best response is simply to set no window and the platform will be seeded with users of type 4, who otherwise have no platform when the original platform set a window of [1,3]. But notice that, under proportional switching, users of type 3 will have compatibility with all of the users on platform 2 and some users who they are not compatible with on platform 1. Thus, users of type 3 will defect to platform 2. But now notice that users of type 2 are compatible with all of the users on platform 2 but not on platform 1, so those users will defect as well. Finally, users of type 1 will stay on platform 1, leaving platform 1 with only an $\varepsilon$-fraction of the population. We can thus construct populations where the first platform can lose an arbitrarily large fraction of its users if another platform joins and sets a different window, for any user bandwidths $\gamma_1, \ldots, \gamma_n$.

When the original platform chose a window to capture the most users, it ultimately created space for another platform to attract users excluded from the first platform and pull a large fraction of users from the first platform to the second one. Notice that, if platform 1 wants to defend against the possible emergence of platform 2, it should set no moderation window and let users of type 1 leave the platform, even though this is suboptimal in the setting where they are the only platform. Thus, just like a platform may need to choose windows achieving smaller communities (compared to what they would choose in a single-platform setting) in order to defend against possible competition.

**Incumbency bias.** For some populations, a platform has significant flexibility in the content moderation policy that it chooses, and has no risk of losing users to other platforms, for any values of $\gamma_1, \ldots, \gamma_n$. Thus, after platform 1 chooses a policy, no users leave platform 1, regardless of which policy it chooses among a large set of possible policies. Consider fig. 5 where $M$ is some large integer and $\theta := \theta_1 = \cdots = \theta_n = (M - 1)/(M + 5)$. There is only one user of types 1-3 and 5-7.

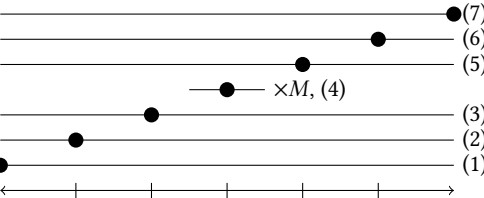

**Figure 5: A population with incumbency bias.** $M$ **is some large integer and** $\theta := \theta_1 = \cdots = \theta_n = (M - 1)/(M + 5)$.

Suppose all users start on platform 1 and that platform 1 chooses any window that includes users of type 4. Notice that the platform will be stable because all users except type 4 will have positive utility on any platform and users of type 4 will have their compatibilities satisfied. Now suppose platform 2 is seeded with whichever users were banned from platform 1 inside platform 2's window. No users from type 4 will defect onto the new platform, since the first user to defect would have negative utility (or at most zero utility if the second platform was empty). Similarly, no users other than type 4 on the platform will defect, since they all derive higher utility from the larger, incumbent platform.

Notice that the first platform is free to choose moderation policies that are sub-optimal, in the sense that they need not capture all of the users on the platform. For example, it could set a window of [2,6], excluding users 1 and 7: perhaps such ideas at the far extremes pose risks to society that the platform does not want support. Alternately, a partisan platform that favors right-leaning or left-leaning perspectives could set a window of [4,7] or [1,4] and thus skew the range of allowable speech without the risk of losing a substantial fraction of its membership. No other users would leave the platform as a result of the decision to disallow the extreme users: if the platform has some preferences over user speech (as long as that preference includes users of type 4), it is free to enforce these preferences on the users without the threat of losing users to the competition. Generally, notice that the first platform can choose from an arbitrarily large number of windows, if we add more users on the left and the right. That is, for a general population, suppose there are $u$ users on either side of the stack of $M$ users in the center of the platform. Then if $\theta = (M-1)/(M+2u-1)$, whichever window the platform sets, as long as it includes the stack of $M$ users in the center, will be the range of allowable speech for the vast majority of users on the platform. All of these choices could have real-world consequences for users' behavior off the platform: if policymakers choose positions of policies based on the range of allowable speech or if individuals make choices based on their perception of the range of socially allowable speech, the platform may have significant influence over public opinion and policy. This is an extreme case where ideological platforms (as defined in Section 4) could set their window equal to their interval and achieve maximal utility with respect to the population, as long as their interval covers the set of $M$ users in the center. More complete exploration of these downstream phenomena are interesting directions for future study.

## B.3 Platforms with different personalization systems.

We close this section by returning to the analysis of how varying personalization affects platforms. In particular, we would like to explore whether greater personalization can be a liability for a platforms competing for users. Using the same construction in the proof of Proposition 5.2, we show that, even if platform 1 starts with a large compatible community and offers better personalization and if platform 2 starts empty and has worse personalization, there may be situations in which platform 2 captures a large fraction (bounded below by a constant factor) of the users on platform 1. In other words, a platform with better personalization may not be better off over the long term, even if they start with all of the users in the largest compatible community on the platform.

**Proposition B.1** (Corollary to Proposition 5.2). *There exist users* $\{l_i, p_i, r_i, b_i\}_{i \in [n]}$ *and two platforms Platform 1 and Platform 2 where:*

*(1) personalization for Platform 1 is no worse than on Platform 2 for every user (i.e., $\lambda_{i,1} \le \lambda_{i,2}$ for all $i \in [n]$),*
*(2) all individuals in a largest compatible community start on Platform 1 and Platform 2 is empty,*

*and at equilibrium, Platform 2 can set a window so at least an $\min\{1/(1+b), b/(1+b)\}$ fraction of users in the largest compatible community are on Platform 2. In fact, this result holds for any fixed choices of $b := b_1 = \cdots = b_n$ and $\lambda := \lambda_1 = \cdots = \lambda_n$ and for some appropriately chosen $\lambda' := \lambda'_1 = \cdots = \lambda'_n$.*

The proof of Proposition B.1 is qualitatively similar to Proposition 5.2 but additionally deals with the dynamics of users choosing between multiple platforms. This tells us that, even factoring in the dynamics of users switching between platforms, a platform with worse personalization can out-compete a platform with better personalization, even if the platform with better personalization starts off with a largest compatible community on the platform. Thus, whether personalization provides a competitive advantage is sensitive to the population in question.

## C HOW ROBUST ARE PLATFORMS TO UNANTICIPATED POPULATION CHANGES?

Up to this point, we have assumed that the population of potential users is known to the platform before it sets its moderation policy. This assumption is sensible if platforms have sufficient data to infer the preferences of their potential user base with high fidelity at the time they are setting a moderation policy. Thus, the model we use throughout the rest of the paper implicitly assumes that user switching occurs on a much faster time scale than platform choices whether to set moderation policies. (Alternately, individuals arrive and leave from the population as independent and identically distributed samples from some underlying distribution, the platform could use the results from Section 3.3 to sustain a community that is a constant fraction of the size of the LCC with high probability.) Thus, if the platform is considering its membership over a large number of user switching decisions before it next re-evaluates its moderation policy, it is sensible to treat the population as fixed. However, in some contexts, the assumption of a fixed population may break down: user bases settling on a platform may occur at the same time as the population of potential users *itself* changes.

In this section, we consider the scenario in which populations may change after the moderation policy is set. One of the main motivating reasons to consider population changes is the presence of adversaries in society: a small minority of users who try to influence the membership of the platform. Adversaries may try, via private messages or off-platform activity, to threaten or harass a user in order to force particular users off of the platform. Alternately, the adversaries themselves may join the platform, disguised as genuine users, and may try to manipulate platform membership by strategically speaking at a point that triggers a cascade of users leaving the platform. We focus on small changes (i.e., the addition or removal of a small, constant number of users) after the moderation policy is chosen and ask how significant their impacts could be on a large platform.

A naive platform that fails to anticipate small changes in the user population can be quite fragile: the arrival or departure of just a constant number of users can lead to a mass exodus off the platform. For example, consider the construction in fig. 6 and assume that all users are on the platform. Notice that the platform is stable: each user is consuming a high enough proportion of content that they like and are willing to stay on the platform. But what if a user joins or leaves the population, i.e., the membership of the original set of individuals changes? For example, what if user 1 leaves the population, or another user whose speech point falls outside of user 4's interval joins the population? In both of these cases, users 1 through 4 would all leave the platform for switching orders where they move first, which would cause a mass exodus: all of the rest of the users on the platform could similarly leave (except for the last user, who will be alone, have zero utility and stay). We will call the arrival or departure of a small, constant number of users from the population a *population shock*, and we do not try to explain why the users join or leave the population, instead treating this as an event determined by factors outside of our utility model of users or the control of the platform.

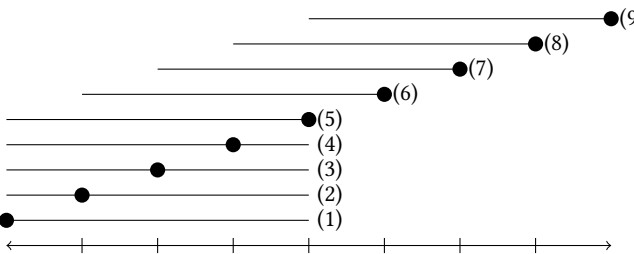

**Figure 6: A platform where it is possible to protect against shocks at a modest cost to size. There are 9 users on the platform, with $\theta_1 = \cdots = \theta_9 = 1/2$.**

A more sophisticated platform might anticipate these potential changes and try to set a moderation policy that is somehow robust to them. Such a platform might be willing to accept a slightly smaller user base in exchange for robustness against a small number of unanticipated arrivals or departures. Returning to the example in fig. 6, note that the if platform sets a narrower window, excluding users 8 and 9, it becomes robust to the arrival or departure of any one user.

This construction is a single example of a class of populations that are robust to population shocks. For any $\theta \in (0, 1)$ and $n$, such a population can be constructed as follows: Each user $i$ will have speech point $i$. The first $\lceil \theta(n-1) \rceil$ users will have intervals $[1, \lceil \theta(n-1) \rceil + 1]$, and each following user $i$ will have interval $[i - \lceil \theta(n-1) \rceil, i]$. The set of initial adopters will be the full population $[n]$. Then, it is easy to show that, to protect against the arrival or departure of any $k$ users, it is only necessary to remove the

$$\left\lceil k \cdot \max\left\{ \frac{1-\theta}{\theta}, 1 \right\} \right\rceil$$

users with the greatest speech points (as long as this is not greater than $\lfloor (1-\theta)(n-1) \rfloor + 1$, which will be true for $n$ large enough). That is, for this family, the platform can sacrifice $O(1)$ users at the time that the moderation policy is set in order to prevent the exodus of $\Omega(n)$ users if a small population shock were to occur.

In these examples, the *price of robustness* is low: the platform could defend against a single change to the population by reducing its user base by $O(1)$. Platforms might hope that this would be true in general: they could exclude $O(1)$ users from the platform when choosing their moderation policy such that a population shock does not cause an exodus of $\Omega(n)$ users.

To make this concept formal in our model, for a population $P$ we will define *a population shock that creates $\tilde{P}$* to mean the addition or removal of individuals from $P$, where $\tilde{P}$ is the population with more or fewer individuals. Then, *a $k$-robust community achievable with a moderation window $I$* is defined as

$$\tilde{s}_k(P, I) := \min_{\tilde{P} \,:\, ||\tilde{P}| - n| \le k} s(\tilde{P}, I);$$

i.e., the size of the largest community under window-based moderation such that if up to $k$ arbitrary users left the population or up to $k$ new individuals joined the population. In Proposition C.1, we show that an $O(1)$ change in the size of the platform does not have implications for $\max_I \tilde{s}_k(P, I)$ in terms of the size of $s_{\text{opt}}(P)$: the tight lower bound on the size of $\max_I s(P, I)$ for Theorem 3.2 also holds for $\max_I \tilde{s}_k(P, I)$, up to an additive constant $k$.

**Proposition C.1** (Corollary to Theorem 3.2). *For a population $P$ where $\theta_{\min} > 1/2$, there exists a moderation window $I \in \mathcal{I}$ such that*

$$\tilde{s}_k(P, I) \ge (2\theta_{\min} - 1)s_{opt}(P) - k.$$

*On the other hand, for all $\theta_{\min} > 1/2$ there exists a population $P$ with population shock $\tilde{P}$ such that for any moderation window $I \in \mathcal{I}$, it holds*

$$\tilde{s}_k(P, I) \le (2\theta_{\min} - 1)s_{opt}(P) + O(1).$$

*and for all $\theta_{\min} \le 1/2$ there exists a population $P$ with population shock $\tilde{P}$ such that for any moderation window $I \in \mathcal{I}$, it holds $\tilde{s}_k(P, I) = O(1)$.*

Intuitively, Proposition C.1 says that the tight lower bound we established in Theorem 3.2 does not get any weaker (besides the size of the shock) for moderation windows under population shocks. This is because the window that achieves the lower bound includes a set of at least $(2\theta_{\min} - 1)$ users with speech points inside with window and intervals that cover the interval. Thus, no users from this set (unless they themselves are removed by the population shock) will leave the platform if any constant number of users

arrive or leave. The second part, establishing the tightness of the bound, also follows from Theorem 3.2, since, for any window $I \in \mathcal{I}$ and $k \ge 0$, it holds

$$\tilde{s}_k(P, I) \le \tilde{s}_0(P, I) \le \max_{I' \in \mathcal{I}} s(P, I').$$

A open question for future research is whether there exists a nontrivial lower bound on the size of $\max_I \tilde{s}_k(P, I)$ in terms of $\max_I s(P, I)$. That is, what is a lower bound on the largest $k$-robust community achievable by a window in terms of the largest community achievable by a window?

## D DYNAMIC WINDOWS AND DEBATE IN A SINGLE DIRECTION

In the body of the paper, we only considered contexts where debate extends in both directions: users will derive disutility at some point from content that is too far to the left *or* the right. But in some contexts, this is not the case; sometimes debate is only sensible in one direction. Consider negotiation about safe levels of average global temperature rise. Every individual would be willing to accept climate change of zero degrees (a left end point of 0), estimates some temperature that they think correctly balances the risks and costs associated with limiting global temperature rise (their speech point), and would be willing to consider propositions of up to some limit (their right end point). We can model this as above by restricting our axis to the nonnegative reals and modeling users with intervals $[0, r_i]$ and speech points somewhere in the interval. Notice that this can also be used to capture populations where intervals are all semi-infinite in one direction, or more generally, where the right-most left endpoint is to the left of the left-most speech point. We could also flip the axis to consider intervals that extend only until some upper limit. This generalizes the user model of Liu et al. [8], where users are always willing to consume content less extreme than theirs. This kind of population with upper-limits might also describe a certain type of hobby-driven community. The hobby, like baking, may allow for increasing levels of intensity in one direction: there are beginner bakers, then bakers relying on more complex techniques, like using sourdough, then hobbyist with even more complex techniques, using only self-milled ancient grains and so on. Each user derives positive utility from content posted to a community focused on the hobby but derives negative utility from content that is too sophisticated. In this community, users agree on their left endpoints, meaning they all derive utility from content from beginners, but they disagree on the level of sophistication that they enjoy.

For such a community, in which disagreement only exists in one direction, we compare the power of dynamic moderation windows with the largest compatible community. We find that in this special case and if $\theta := \theta_1 = \cdots = \theta_n$, the community achievable with dynamic window-based moderation is as large as the largest compatible community and can be found in polynomial time. Note that this is a much stronger bound than Theorem 3.2, which holds for general populations. Whereas Theorem 3.2 says that a window can only always hope to achieve a $(2\theta - 1)$ fraction of an LCC on a general population, in this case, a dynamic window can capture a complete LCC for all $\theta$.

First, we formally define dynamic moderation windows. A *dynamic moderation window* $I : \mathbb{Z}_{>0} \rightarrow \mathcal{I}$ defines a moderation window at each time step. The window at time $t$ will be denoted $I(t) := [l^{(t)}, r^{(t)}]$. In this section, we will refer to non-dyanmic moderation windows as static.

**Proposition D.1.** *For any population $P$ such that $0 = l_1 = \cdots = l_n$ (i.e., all left endpoints are zero) and $\theta := \theta_1 = \cdots = \theta_n$, there exists a dynamic moderation window $I(\cdot)$ such that*

$$s(P, I(\cdot)) = s_{opt}(P)$$

*Further, for all static moderation windows $J \in \mathcal{I}$, there exists a population $P$ such that $0 = l_1 = \cdots = l_n$ (i.e., all left endpoints are zero) and $\theta := \theta_1 = \cdots = \theta_n$ and*

$$s(P, J) \leq (2\theta - 1)s(P, I).$$

Intuitively, the result says that the best dynamic moderation window can always achieve a community as large as the largest compatible community, but a static moderation window cannot.

**Proof of Proposition D.1.** As in the proof of Theorem 3.1, we first construct a maximal compatible set. Then we show how a window can be set so that all users in the moderation window will remain when switching reaches an equilibrium.

**Lemma D.2.** *When the left endpoints of the intervals are the same, the maximal compatible set can be found using the following algorithm.*

---

**ALGORITHM 1:** Maximal compatible set for one-sided intervals.

**Input:** Users $\left\{(l_i, p_i, r_i)\right\}_{i=1}^{n}$ with $l_1 = \cdots = l_n = 0$ and
$\qquad \theta := \theta_1 = \cdots = \theta_n$.
**Output:** A maximal compatible set.
**for** $j \in [n]$ **do**
$\quad$ Let $\mathcal{S} := \left\{i \in [n] : p_i \in I_j, r_i > r_j\right\}$ be the index set of users
$\qquad$ whose speech points are in user $j$'s interval and whose intervals
$\qquad$ are longer than user $j$'s interval.
$\quad$ Let $\mathcal{T} := \left\{i \in [n] : p_i > r_j\right\}$ be the index set of users whose
$\qquad$ speech points are outside of user $j$'s interval.
$\quad$ Let $\mathcal{W} := \left\{j\right\} \cup \mathcal{S}$ be a working set.
$\quad$ Add $(1/\theta - 1)|\mathcal{S}|$ arbitrary intervals from $\mathcal{T}$ to $\mathcal{W}$.
$\quad$ Check if $\mathcal{W}$ is the largest compatible set found so far. If so, record
$\qquad$ its entries.
**end**

---

We can interpret this lemma as a way of understanding the mechanisms by which an agreement is formed: in any stable community, there must exist some user (call them user $i$) with the shortest interval in any compatible set. This user must be willing to stay on the platform, so they must have nonnegative utility (or, equivalently, a $\theta_i$ fraction of compatible content) from other users. All other users on the platform have intervals that contain user $i$'s interval, so they must derive at least as much utility from the platform as user $i$. Given that the user with the shortest interval is willing to to stay, all users with longer intervals must also be willing to stay.

**Proof of Lemma D.2.** We will prove that the algorithm above finds a set of equivalent or greater size to an any compatible set.

Let $\mathcal{W}, \mathcal{S}$ and $\mathcal{T}$ be those recorded by the algorithm. Let $\mathcal{W}'$ be an arbitrary compatible set, let $j$ be index of the shortest interval in $\mathcal{W}'$, let $\mathcal{S}'$ be the set of intervals in $\mathcal{W}'$ compatible with $j$ and let $\mathcal{T}'$ be the set of intervals in $\mathcal{W}'$ that are not compatible with $j$.

Notice $\mathcal{S}' \subseteq \mathcal{S}$ and $|\mathcal{T}'| \leq \mathcal{T}$ since the algorithm had $\mathcal{S}'$ in iteration $j$. Notice that intervals in $\mathcal{S}'$ are all mutually compatible, since $p_i < r_j$ and $r_i > r_j$ for all $i \in \mathcal{S}'$. This also implies that any user in $\mathcal{S}'$ will still have their compatibility constraints satisfied if up to $(1/\theta - 1)|\mathcal{S}'|$ intervals join the platform. Notice that we can have a maximum of $(1/\theta - 1)|\mathcal{S}'|$ intervals from $\mathcal{T}$ in $\mathcal{W}$ or $\mathcal{W}'$; any more than $(1/\theta - 1)|\mathcal{S}'|$ intervals added to the working set would violate the compatibility constraint of interval $j$. Also, any user in $\mathcal{T}'$ will have their compatibility constraints satisfied if up to $(1/\theta - 1)|\mathcal{S}'|$ are added to the set. This is because all intervals in $\{j\} \cup \mathcal{S}'$ are compatible with all intervals in $\mathcal{T}'$. To see this, note $p_i < r_j$ for all $i \in \mathcal{S}'$ and $p_i > r_j$ for all $i \in \mathcal{T}'$. Trivially $r_i > p_i$, so $r_i > p_k$ for all $i \in \mathcal{T}'$ and $k \in \mathcal{S}'$. We also assumed $l_1 = \ldots l_n$, so $p_k \in I_i$ for all $i \in \mathcal{T}'$ and $k \in \mathcal{S}'$.

Since $|\mathcal{S}'| \leq |\mathcal{S}|$ and we could proceed by exchanging elements from $\mathcal{T}' \setminus \mathcal{W}$ with elements from $\mathcal{W} \setminus \mathcal{T}'$, we know $\mathcal{W}$ is at least as large as $\mathcal{W}'$. □

Next, we want to show that a dynamic window can be chosen such that the resulting set, for any starting orientation and switching order, is as large as the LCC.

Our dynamic window will be as follows. Let $\mathcal{S}^*$ be a set of users in the LCC whose content are compatible with the user with shortest interval, call it $j^*$. Let $\mathcal{W}^*$ be the set of users in the largest compatible community, where we take the union over $\mathcal{S}^*$ and the $(1/\theta - 1)|\mathcal{S}^*|$ users with the least speech points greater than $r_{j^*}$. We set our dynamic window to be the minimum length interval that covers $\mathcal{S}^*$ until all members of $\mathcal{S}^*$ are on the platform, at which point we set it to be the minimum length interval that covers $\mathcal{W}^*$. Notice that any user from $\mathcal{S}^*$ will join the platform under a window that covers $\mathcal{S}^*$, since they will derive positive utility from all users allowed on the platform. Also notice that after all users from $\mathcal{S}^*$ join the platform, there are only $(1/\theta - 1)|\mathcal{S}^*|$ users outside of user $j^*$'s interval allowed on the platform. Each of these users will join the platform when given the chance, since they have all of the compatibilities of user $j^*$ (and possibly more) and user $j^*$ has at least $|\mathcal{S}^*|$ compatibilities and no more than $(1/\theta - 1)|\mathcal{S}^*|$ incompatibilities. Thus, user $j^*$ is satisfied. And since all users in $\mathcal{S}^*$ have intervals longer than that of $j^*$, they must also derive positive utility from the platform. □

# E  HETEROGENEOUS SPEECH FREQUENCIES

Consider adding an extra parameter to each of the users on the platform: speech frequency. That is, some users will speak more frequently than others, and thus will show up more often in the feeds of other users and therefore exert more influence on whether users are willing to participate in the platform. User $i$'s speech frequency will be denoted $f_i \geq 0$. We will call these *variable-frequency* populations and the normal version of the model *fixed-frequency*. We show that the maximal compatible set problem on the variable-frequency problem can be reduced to the fixed-frequency problem in polynomial time in the total amount of speech. For simplicity, we will just consider integer values of $f_i$, but the analysis can be extended to rationals (or arbitrarily tight approximations of irrationals) by multiplying all frequencies by a constant such that all

frequencies are integers. In this problem, each user can produce content at a rate between zero and $f_i$ and will leave the platform if the total amount of speech compatible with them (weighted by frequencies) is not at least a $\theta_i$ proportion.

**Proposition E.1.** *For a variable-frequency population $Q = \{l_i, p_i, r_i, \theta_i, f_i\}_{i \in [n]}, S_0$ let $s_{opt}(Q)$ be the LCC of users that a platform could sustain if it could choose an arbitrary set of users to allow on the platform and set caps on the frequencies of user speech.*

*Define $f := \sum_{i=1}^{n} f_i$. Then*

$$s_{opt}(Q) = s_{opt}\left(\left\{I_i, p_i, \theta_i + \frac{f_i - 1}{f - 1}\right\}_{i \in [n]}, S_0\right). \quad (2)$$

In other words, Proposition E.1 says that the largest compatible community under a variable-frequency problem can be computed by finding the largest compatible community of a fixed-frequency problem in polynomial time in the sum of frequencies.

**Proof of Proposition E.1.** For a given variable-frequency population, we create a construction of a fixed-frequency population (where users all have the same frequency) from the variable-frequency problem. Start by creating $f_j$ identical users with the same $r_j, p_j$ and $l_j$. We will choose their tolerance threshold in the fixed-frequency problem so that it is equivalent to the threshold in the variable-frequency problem. Notice that, since we created $f_j$ identical users in the fixed-frequency problem, each of the users are compatible with each other. However, since users do not get utility from consuming their own speech, we need to increase user thresholds in the fixed-frequency problem to account for the "free" utility users get from listening to other identical users. Define

$$\theta'_j := \theta_j + \frac{f_j - 1}{f - 1}$$

and notice that users in the variable-frequency problem are satisfied if and only if each user is satisfied in the fixed-frequency problem. To see this, suppose a user $j$ in the variable-frequency problem is satisfied. Then at least $\theta_j$ proportion of the speech in the variable-frequency problem is compatible with user $j$. Let $j_l$ for $l = 1, \ldots, f_j$ be the users identical to user $j$ in the fixed-frequency problem. Then at least

$$\frac{f_j - 1 + \sum_{k \neq j} f_k \mathbb{1}\{p_k \in I_j\}}{f - 1} \geq \theta_j + \frac{f_j - 1}{f - 1}$$

speech in the fixed-frequency problem are compatible with user $j_l$. Conversely, if user $j_l$ is satisfied in the fixed-frequency problem, then at least $\theta'$ proportion of speech is compatible. Then the same holds for all other $j_k$ for $k = 1, \ldots, f_j$ since $j_l$ and $j_k$ have the same interval and tolerance threshold.

Then user $j$ in the variable-frequency problem has at least

$$\theta'_j - \frac{f_j - 1}{f - 1} = \theta_j$$

speech they find compatible. $\square$

# F PLATFORM STABILITY AND SWITCHING DYNAMICS

In both our single platform model and our multiple platform model, platforms may never reach an equilibrium. However, under some conditions, such as when all compatibilities are mutual (i.e., if user $i$ is compatible with $j$, then $j$ is compatible with $i$) and participation thresholds are homogeneous (i.e., $\theta_1 = \cdots = \theta_n$), then the platform is guaranteed to reach an equilibrium — for either the single platform case or multiple platforms. We prove these results below.

## F.1 The single platform case

Our first observation is that switching dynamics need not stabilize and may not have any stable arrangements of users where the platform is nonempty. After that, we show that when all compatibilities are mutual and participation thresholds are homogeneous, then the platform is guaranteed to stabilize.

To show that some populations may never stabilize under any switching order, we use the following example.

**Example F.1.** *Suppose we have a population of n users arranged as in fig. 7 with the number of users for each type to the right along with group numbers in parentheses. Let $\theta_i = 2/3$ and $\lambda_i = 1$ for all $i \in [n]$. Further, suppose there is no moderation window.*

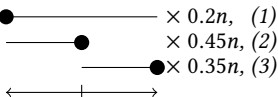

$$\begin{array}{ll}
\times 0.2n, & (1) \\
\times 0.45n, & (2) \\
\times 0.35n, & (3)
\end{array}$$

**Figure 7: Platform cycling. Suppose $\theta_i = 2/3$ for all $i$.**

**Proposition F.2.** *There are no equilibria in Example F.1.*

**Proof of Proposition F.2.** Notice that regardless of which users the platform is seeded with, all the users of type 1 will always choose to join, since they are compatible with all possible users on the platform. To see this, observe that when a user of type 1 is next in the switching order, they always will join a nonempty platform and will never leave, since every speech point is within their interval and this means they can have a minimum of 0 utility. Additionally, once a platform has a nonempty set of seed users, it will never become empty. This is because users leave one at a time, so the last user on the platform will have zero utility and will thus not leave the platform. Thus, we will start our analysis assuming all members of type 1 have joined. Also notice that if any one user of type 1, 2 or 3 is on the platform and derives positive utility, all of the users must, so any equilibrium will have all users of a given type or none. This establishes the following equivalence between equilibria in our example for any $n$ and the $\theta_i$ are as above. (Throughout, when we present examples, they will generally apply regardless of whether the platform is small or large.) The ratios between the number of users are the crux of the example.

We proceed to the proof that the construction above results in cycling. Any nonempty platform without 1 is not an equilibrium, since users 1 will always join the platform. The platform with just 1 is not an equilibrium since users 2 will want to join. The platform with 1 and 2 or 1 and 3 are not equilibria because in each case users 3 will want to switch from their current status. This is because, in the platform with 1 and 2, the users 3 will each have at least $0.45n/(0.2n + 0.45n) > 2/3$ content they are compatible with, so those users would join the platform if given the chance. In the platform with 1 and 3, users 3 would have only $0.35n/(0.35n + 0.2n) < 2/3$ content they are compatible with, so those users would

leave the platform if given the chance. The platform with 1, 2, and 3 is not an equilibrium because in this case user 2 wants to leave, since they would have a proportion of only 0.65 of content they like.

This process is easiest to visualize when users from each group move *en masse*. Suppose users 1 start on the platform. Users 2 join, since they derive positive utility from other members of their group and users 1. Then users 3 join, since they derive positive utility from users 2 and themselves. But this causes users 2 to leave, and the process repeats.                                                     □

Cycling does not occur, however, when the population consists of users where all compatibilities are mutual and users have the same participation thresholds.

**Proposition F.3.** *Consider a population where, if user $i$ is compatible with user $j$, then user $j$ is compatible with user $i$, where $b := b_1 = \cdots = b_n$ and $\lambda := \lambda_1 = \cdots = \lambda_n$. Then, for any switching order, the platform will reach an equilibrium.*

**Proof of Proposition F.3.** We will show that every finitely often, the sum of the current utilities of individuals increases by at least a constant amount. Once we show this, since the maximum utility is bounded by $n(n-1)$ (this is achieved in the extreme case that all users are mutually compatible) and each switch increases total utility by at least a constant amount, the switching process must converge in finite time.

First, observe that when a user joins a platform, it must be because their utility for participating on the platform is nonnegative. As before, let us define $u_i(\mathcal{S})$ to be the utility of user $i \in [n]$ if they were to join the platform and $\mathcal{S}$ to be the index set of users currently on the platform. Also, define $v_i(\mathcal{S})$ to be the sum of utilities that all users on the platform would derive from user $i$. That is,

$$v_i(\mathcal{S}) = \sum_{\ell \in \mathcal{S}\setminus\{i\}} \mathbb{1}\{p_i \in I_\ell\} - \lambda b \mathbb{1}\{p_i \notin I_\ell\}.$$

Notice, since all compatibilities are mutual, $v_i(\mathcal{S}) = u_i(\mathcal{S})$ for all $i, j$. Then, since user $i$ is joining the platform, $u_i(\mathcal{S}) \geq 0$ and thus $v_i(\mathcal{S}) \geq 0$. That is, user $i$ derives nonnegative utility from joining the platform, and the current members of the platform derive nonnegative aggregate utility from user $i$. Thus, every time a user joins the platform, utility among the set of users on the platform is nondecreasing.

Similarly, when a user leaves a platform, it must be because their utility for participating was strictly negative. Thus, every time a user leaves the platform, net utility of users on the platform must strictly increase.

The utilities of those who are not currently on the platform do not change, so the sum of the current utilities of all users must not decrease every time a user joins a platform and must strictly increase every time a user leaves a platform. Since there are only $n$ users, only $O(n)$ switches can occur before total utility strictly increases.

Next, we argue that if total utility increases, it does so by at least a constant in the parameters of the population. Notice that $v_i(\mathcal{S})$ must be an integer multiple of 1 and $\lambda b$ where each integer is less than the size of the platform. The set

$$\left\{ x - y\lambda b \mid x - y\lambda b > 0, x \in [n], y \in [n] \right\}$$

is finite, so it must have achieve a positive minimum that is a function of $b$ and $\lambda$.

Next, we prove corresponding results for cycling on multiple platforms.

## F.2 Cycling with multiple platforms

As with the single-platform case, platforms under competition can cycle indefinitely, regardless of whether the platforms are large (greater than the size of user bandwidths) or small (lesser than the size of user bandwidths). We demonstrate this through several examples, and then we prove a result similar to Proposition F.3 which gives conditions under which an equilibrium is guaranteed for multiple platforms.

We will construct examples where users can cycle between platforms, regardless of whether they are following proportion- or utility-based switching. Recall that users follow proportion-based switching when the amount of content the platform might serve them is larger than their bandwidth and utility-based switching when it is smaller. With no personalization, the amount of content the platform might serve them is equal to the number of users on the platform; with personalization, some content outside of a user interval is filtered out, reducing the amount of content for a given user.

In fig. 8, there are four types of users. Suppose they each have $\theta_i = 7/9$ and $\lambda_i = 1$ for all $i \in [n]$. Two of the types of users, types 1 and 4, have intervals that cover all other user speech. Meanwhile, users of types 2 and 3 only derive positive utility from some of the other users. Users of type 2 only derive utility from users of type 1, 2 and 4, while users of type 3 only derive utility from users of type 2 or 3. Users of type 1, 2 and 3 start on platform 1, while users of type 4 start on platform 2. Further, suppose neither platform sets a moderation window.

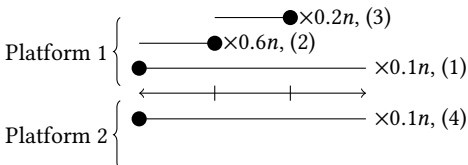

**Figure 8: Platform cycling under proportional- and utility-based switching. Fix $\theta := \theta_1 = \cdots = \theta_n = 7/9$ and $\lambda := \lambda_1 = \cdots = \lambda_n = 1$. The first platform is displayed above the axis and the second platform is displayed below the axis.**

We will argue that, under both utility-based and proportional-based switching, for some starvation-free switching order, the platform will never reach an equilibrium.

We will specify switching orders in which users of a given type move sequentially: all users of type $j$ will be given a chance to switch in a row for each $j$. For proportional-based switching, we choose an order of types $1, 2, 3, 4, 1, 2, 3, 4, \ldots$. Observe that users of type 1 and 4 will never leave their platform, since they each derive positive utility from every other user on the platform. Thus, when users of type 1 are first in the switching order, they stay. Next in the order is users of type 2. Any user of type 2 is completely compatible with the users on platform 2 (which are users of type 4), and so

will switch to the second platform. All of the rest of users of type 2 will follow. Now, users of type 3 will have negative utility on platform 1 but positive utility on platform 2, leading each of them to switch to platform 2. Finally, users 4 will not leave their platform since they derive positive utility from every other user. Then, the process repeats, users 1 stay on the platform, users 2 switch, then users 3 switch and users 4 stay. This gets us back to the starting arrangement and thus the platforms will cycle indefinitely.

For utility-based switching, we choose 1, 2, 3, 4, 2, 3, 1, 2, 3, 4, 2, 3, ... for our switching order, such that all users of type 1 are allowed to switch, then type 2, then 3, then 4, then 2, then 3 then the whole sequence is repeated. We will set each user $i$'s bandwidth to $\gamma_i = n$: they could consume all possible content if all users were all on one platform. When users of type 1 are up in the switching order, they derive positive utility from every user so will choose whichever platform is larger. This is platform 1, so they will stay. Users of type 2 are next in the switching order. The first user of type 2 in the switching order will have total utility $(0.7n - 1) - 0.2nb$ on platform 1 (the minus one term comes from the fact that users do not derive utility from themselves), and utility $0.1n$ utility on platform 2. It can trivially be verified that for $b = 7/2$, it holds that $(0.7n - 1) - 0.2nb < 0.1n$, for all $n \geq 1$ and so the first user of type 2 switches to platform 2. But then each subsequent user of type 2 has even lower utility on platform 1 and greater utility on platform 2. Thus, they all switch to platform 2. Next, users of type 3 have negative utility on platform 1 and positive utility on platform 2, so they also all switch to platform 2. Finally, users of type 4, like users of type 1, will always go to the platform with the larger number of users, which is platform 2 when they are given the chance to switch, so they will stay on the platform. But now the platforms are symmetric to before switching started, so users of type 2 and 3 switch back to platform 1 and the process repeats, cycling indefinitely.

Notice that under either type of switching, users of type 1 will stay on platform 1, then users of type 2 will leave for platform 2, then user 3 will follow users of type 2 to platform 2 and so on. Thus, for general populations with two platforms, platforms can cycle when platforms are below and above their bandwidth thresholds. However, for utility-based switching with mutually compatible users and homogeneous utility functions, this is not the case. We show this in the next result, which closely follows the logic of Proposition F.3.

**Proposition F.4.** *Consider a population where, if user $i$ is compatible with user $j$, then user $j$ is compatible with user $i$, where $b := b_1 = \cdots = b_n$ and $\lambda := \lambda_1 = \cdots = \lambda_n$. Suppose user bandwidths $\gamma_i \geq n$ are no smaller than the size of the population. Then, for any switching order, the platform will reach an equilibrium.*

**Proof of Proposition F.4.** The logic of the proof of Proposition F.4 closely follows Proposition F.3. As before, we will show that total utility always strictly increases every finitely many switches. Then, since utility is bounded from above and every time utility increases, it must increase by at least a constant amount in the population parameters, this shows that the platform must converge to an equilibrium.

Define $u_i(\mathcal{S}_j)$ to be the utility that user $i$ would derive on platform $j$ where $\mathcal{S}_j$ is the current set of users on platform $j$. And let $v_i(\mathcal{S}_j)$ be the utility derived from user $i$'s on platform $j$. That is,

$$v_i(\mathcal{S}_j) = \sum_{\ell \in \mathcal{S}_j \setminus \{i\}} \mathbb{1}\{p_i \in [l_\ell, r_\ell]\} - \lambda b \mathbb{1}\{p_i \notin [l_\ell, r_\ell]\}$$

Now, notice that $u_j(i) = v_j(i)$ and thus, since when a user switches or leaves platforms, total utility must increase. Since there are only finitely many values of $x - y\lambda b > 0$ for $x, y$ nonnegative integers no greater than $n$ and since total utility is bounded by $n(n - 1)$, the platforms must converge. □

# G DEFERRED PROOFS

In what follows, we restate each result before the proof for convenience.

## G.1 Proofs for Section 3

**Theorem 3.1.** *For any population $P$ where $\theta_1 = \cdots = \theta_n = 1$, there exists a moderation window $I^* \in \mathcal{I}$ such that*

$$s(P, I^*) = s_{opt}(P).$$

*Moreover, such a window $I^*$ can be identified in $O(n^3)$ time.*

**Proof of Theorem 3.1.** We first need to find the size of the largest compatible community. We do this by construction in Lemma G.1.

**Lemma G.1.** *When $\theta_1 = \cdots = \theta_n = 1$, the largest compatible community can be found using algorithm 2.*

---

**ALGORITHM 2:** Largest compatible community for $\theta_1 = \ldots \theta_n = 1$

**Input:** *Users $\left\{(l_i, p_i, r_i)\right\}_{i=1}^{n}$ with participation thresholds $\theta := \theta_1 = \cdots = \theta_n = 1$ and set of initial adopters $\mathcal{S}_0$.*
**Output:** *The largest compatible community.*
*Relabel intervals so $p_1 \leq p_2 \leq \cdots \leq p_n$.*
**for** *each $i, j \in [n] \times [n]$ such that $i, j$ mutually compatible and $i < j$* **do**
    *Initialize some set $\mathcal{S}_{i,j} \leftarrow \varnothing$.*
    *For each $k \in [n]$ such that $i < k < j$, add $k$ to $\mathcal{S}_{i,j}$ if and only if it is mutually compatible with both $i, j$.*
    *Check if $\mathcal{S}_{i,j}$ is the largest compatible community found so far. If so, record its entries.*
**end**
*Return the largest $\mathcal{S}_{i,j}$.*

---

**Proof of Lemma G.1.** Notice that, when $\theta_j = 1$ for all $j$, every user in a compatible community must be compatible with every other user. Equivalently, every compatible community must have the property that every user interval in the set covers all speech points in the set. Each compatible community must have a maximum and minimum speech point, and we iterate over these choices of speech points to find the largest one. In the for-loop, for given pair of users $i$ and $j$, we enforce the properties that all other speech points fall in $[p_i, p_j]$ and all intervals in the compatible community cover $[p_i, p_j]$. To see the first part, notice that the speech points are sorted and we only search over $k$ such that $i < k < j$ in the inner for-loop. To see the second part, notice that we are requiring that $k$ be mutually compatible with $i$ and with $j$, which means that user $k$'s interval extends left at least until $p_i$ and right until $p_j$. Since this means that every user interval in the set covers all speech points,

this proves that each set created in an iterator of the for-loop is compatible. Since we iterate over all choices of $p_{\min}$ and $p_{\max}$, we must find every compatible community, which means that we can take the largest of these as the maximum. □

We now proceed with the rest of the proof. Suppose we choose the moderation window to be equal to the minimum and maximum speech points in the largest compatible community. Notice that every user in the largest compatible community covers the entire window, and this means that no user can join the platform whose speech is not compatible with them. Thus, every user in the largest compatible community will have their participation threshold satisfied and will stay on the platform. Now we just need to show that every user in the moderation window will join the platform. Since they cover the whole window and no users whose speech falls outside the window are allowed on, every user in the largest stable platform will gain positive utility from every other user on the platform at any time in the process. So users in the largest stable set will always choose to join the platform when it is their turn to choose, and they will never leave. Further, these users are the only ones whose intervals cover the whole range from $p_{\min}$ to $p_{\max}$, so no other users will join the platform and stay forever. □

**Theorem 3.2.** *For any population $P$ where $\theta_{\min} > 1/2$, there exists a moderation window $I \in \mathcal{I}$ such that*

$$s(P, I) \geq (2\theta_{\min} - 1)s_{opt}(P)$$

*that can be identified in $O(n^3)$ time. On the other hand, for all $\theta_{\min} > 1/2$, there exists a population $P$ such that for any moderation window $I \in \mathcal{I}$, it holds*

$$s(P, I) \leq (2\theta_{\min} - 1)s_{opt}(P) + O(1).$$

*and, for all $\theta_{\min} \leq 1/2$, there exists a population $P$ such that for any moderation window $I \in \mathcal{I}$, it holds $s(P, I) = O(1)$.*

**Proof of Theorem 3.2.** For the first part of the theorem, we start with a result showing that any compatible community $\mathcal{S}$ for $\theta_{\min} \geq 1/2$ must include a set of users that are compatible with all other users in the compatible community, proportional to the size of $\mathcal{S}$.

**Lemma G.2.** *Let users $\left\{(l_i, p_i, r_i)\right\}_{i=1}^{n}$ for $\theta_{\min} \geq 1/2$ contain a compatible community of at least $k$ users. Then there must exist at least $k(2\theta_{\min} - 1)$ users which are compatible with all other users in the set.*

**Proof of Lemma G.2.** Suppose the users in the compatible community are ordered so that $p_1 \leq p_2 \leq \cdots \leq p_k$ and define $\theta := \lceil (k-1)\theta_{\min} \rceil / (k-1)$. We will show users $(1-\theta)(k-1) + 1$, $\ldots$, $\theta(k-1) + 1$ are compatible with every other user. To see why this is true, consider an arbitrary user $j$. Since the set is compatible, user $j$ must be compatible with at least $\theta(k-1)$ other users. If $j < \theta(k-1) + 1$, then user $\theta(k-1) + 1$ must be compatible with user $j$ since there are not $\theta(k-1)$ users $i$ such that $p_i < p_{\theta(k-1)+1}$ since we labeled the users in ascending order of speech point. Similarly, if $j \geq \theta(k-1) + 1$, then user $\theta(k-1) + 1$ must be compatible with user $j$ since there are not $\theta(k-1)$ users $i$ such that $p_i > p_{\theta(k-1)+1}$. The same reasoning will show that user $(1-\theta)(k-1) + 1$ is compatible with user $j$: If $j > (1-\theta)(k-1) + 1$, then user $(1-\theta)(k-1) + 1$ must be compatible with user $j$ since there are not $\theta(k-1)$ users $i$

such that $p_i > p_{(1-\theta)(k-1)+1}$ since we labeled the users in ascending order of speech point. Similarly, if $j \leq (1-\theta)(k-1) + 1$, then user $(1-\theta)(k-1) + 1$ must be compatible with user $j$ since there are not $\theta(k-1)$ users $i$ such that $p_i < p_{(1-\theta)(k-1)+1}$. □

By Lemma G.2, there must be a set $\mathcal{S}$ of mutually compatible users where $|\mathcal{S}| \geq (2\theta_{\min} - 1)s_{opt}(P)$. We will construct an window such that all users in $\mathcal{S}$ are willing to use the platform regardless of switching order. Let $p_{\min}$ and $p_{\max}$ be the left- and right-most speech points in $S$, i.e.,

$$p_{\min} \triangleq \min_{i \in S} p_i$$

$$p_{\max} \triangleq \max_{i \in S} p_i$$

Because all of the users in $\mathcal{S}$ are mutually compatible, their intervals must all include both $p_{\min}$ and $p_{\max}$. We will chose our window to be exactly $[p_{\min}, p_{\max}]$, so that every user in $\mathcal{S}$ covers the entire window. Next, we will argue that when it is the turn of any user from $\mathcal{S}$ in the starvation-free switching order, they will join the platform. To see this, we will first argue that the platform is nonempty when the first user from the set is given the opportunity to join. The platform starts with some user, the first in the switching order whose speech falls in the interval. But since switching happens one user at a time, at least one user whose speech is inside the window will be on the platform at all times: a single user will never leave, since they derive zero utility. Since every user allowed on the platform is compatible with every interval in $\mathcal{S}$, the first user from the set that is allowed to join will do so. But then since their interval covers the whole window, they will derive positive utility from any user on the platform, so that user will never leave. And this means that every user from the mutually compatible community will join the platform eventually. Finally, since there are greater than $(2\theta_{\min} - 1)s_{opt}(P)$ speech points in $[p_{\min}, p_{\max}]$ that cover the whole interval by the existance of $s_{opt}(P)$, and all of them will join the platform when given the chance, at least this number will join the platform and never leave.

For upper bound in the theorem, we will use constructions where all individuals have the same participation threshold: $\theta = \theta_1 = \cdots = \theta_n$. We will handle the case when $\theta \in (1/2, 1)$ first and the one for $\theta \leq 1/2$ second. The first case will have an upper bound for a platform with window-based moderation of $(2\theta - 1)s_{opt}(P) + O(1)$ and the second will have an upper bound of no more than a constant number of users.

For the first construction, without loss of generality, let $(\theta - 1/2)n$ be an integer. (Otherwise, simply define a new $\theta' := \lfloor (\theta - 1/2)n \rfloor / n + 1/2$ and substitute $\theta'$ throughout instead.) Let there be

$$K := 2\left\lceil \frac{(1-\theta)n - 1}{\left(\theta - \frac{1}{2}\right)n} \right\rceil$$

sets of $Q \approx (\theta - 1/2)n$ users (some stacks will have one fewer user so that the total number of users in these $K$ stacks adds up to $2((1-\theta)n - 1)$). Users within a set will have identical speech points and intervals. We will specify $n$ later. For $j = 1, \ldots, K/2$ let the $j$th set of users have speech point $j$ and semi-infinite intervals $[j, \infty)$ (or equivalently, finite intervals that extend past the right-most speech point in the construction). For $j = K/2 + 3, \ldots, K$, let the $j$th set of users have speech point $j + 3$ and semi-infinite intervals

$(-\infty, j+3]$. Finally, let there be two additional sets of users: the first will have speech point $K/2 + 1$ and interval $[K/2 + 1, \infty)$ and the second will have speech point $K/2 + 2$ and interval $(-\infty, K/2 + 2]$. Each of these sets will have $(\theta - 1/2)n + 1$ users. The set of initial adopters will be $\mathcal{S}_0 = \varnothing$. A diagram of the construction is shown in fig. 9.

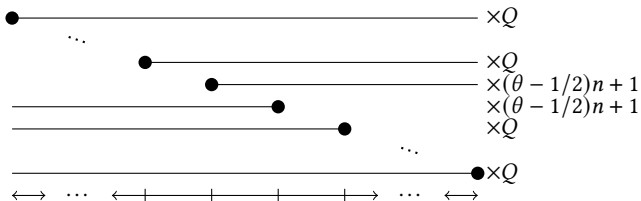

**Figure 9: A construction for $\theta > 1/2$. A window can achieve no more than a $(2\theta - 1)n + 2$-sized platform.**

Now we will argue that, for any window, there exist a stable platform with no more than $(2\theta - 1)n + 2$ users. Let the window be $[L, R]$ where the endpoints are integers in $\{1, \ldots, K + 3\}$ without loss of generality. Then, consider switching orders in which the sets with speech points at $L$ are given the opportunity to join first and the speech points at $R$ are given the opportunity to join next. Observe that all users in each set will join the platform, since they are compatible with all users on the platform already.

Next, we will argue that no other users will join the platform. First notice that all users inside the window still on the platform are compatible with exactly one of the set with speech point $L$ or the speech point $R$. Let $N_L$ and $N_R$ be the number of users in the sets of users with speech point at $L$ and with $R$, respectively. Then, notice that if $N_L/(N_L + N_R) < \theta$ and $N_R/(N_L + N_R) < \theta$, then no other users will join the platform. The conditions in the previous sentence are equivalent to the condition

$$\frac{1 - \theta}{\theta} N_L < N_R < \frac{\theta}{1 - \theta} N_L.$$

Now notice that sets that are on the periphery of the set (i.e., the $K$ sets of $Q$ users) have sets that are no more than size 1 difference. Thus, the conditions are trivially satisfied for large enough $n$ for windows $[L, R]$ where $L$ and $R$ are both the speech points of users in the $K$ sets of $Q$ users. If either $L$ or $R$ indexes one of the core sets with $(\theta - 1/2)n + 1$ users, then we just need to verify the inequality where, without loss of generality, $L$ is the speech point of the core set of users. Thus, the inequalities we want to prove are

$$\frac{1 - \theta}{\theta} \left( \left( \theta - \frac{1}{2} \right) n + 1 \right) < Q < \frac{\theta}{1 - \theta} \left( \left( \theta - \frac{1}{2} \right) n + 1 \right).$$

For the first inequality, we can notice

$$Q \geq \left( \theta - \frac{1}{2} \right) n - 1$$

$$> \frac{1 - \theta}{\theta} \left( \left( \theta - \frac{1}{2} \right) n + 1 \right). \qquad \text{(since } (1 - \theta)/\theta < 1)$$

for large enough $n$. The second inequality is satisfied by noticing

$$Q \leq \left( \theta - \frac{1}{2} \right) n$$

$$< \frac{\theta}{1 - \theta} \left( \left( \theta - \frac{1}{2} \right) n + 1 \right) \qquad \text{(since } 1 < \theta/(1 - \theta))$$

for large enough $n$. Finally, since $Q \leq (\theta - 1/2)n$, no window can capture more than $(2\theta - 1)n + 2$ users (which is achieved by selecting the two center stacks of users). This concludes the proof for this construction.

Next we handle the case when $\theta \leq 1/2$. We will start by describing the unique largest compatible set (in terms of some $s_{\text{opt}}(P)$ to be specified later), which will consist of users indexed $1, \ldots, s_{\text{opt}}(P)$. For all $i = 1, \ldots, s_{\text{opt}}(P)$, let $p_i = i$. For $i = 1, \ldots, \lceil \theta(s_{\text{opt}}(P) - 1) \rceil$, let $l_i = i$ and $r_i = i + \lceil \theta(s_{\text{opt}}(P) - 1) \rceil$. For $i = \lceil \theta(s_{\text{opt}}(P) - 1) \rceil + 1, \ldots, s_{\text{opt}}(P)$, let $l_i = i - \lceil \theta(s_{\text{opt}}(P) - 1) \rceil$ and $r_i = i$. Notice that this set is compatible.

Next, we will describe a set of users whose speech is interspersed with the largest compatible community in such a way that, for some switching order, only a constant number of users will ever be on the platform. We will call these users *spoilers*. Spoilers will be constructed differently depending on whether $\theta \leq 1/3$ or $\theta \in (1/3, 1/2)$. The $\theta \leq 1/3$ condition is simpler and illustrates the main intuition, so we start with that first.

Define $M > (1 - \theta)/\theta^2$ to be the number of spoilers per user. For each user $i \in [s_{\text{opt}}(P)]$, each of the $M$ spoilers will be indexed $i_j$ for $j = 1, \ldots, M$, and say these are the spoilers *corresponding* to user $i$. For all $i \in [s_{\text{opt}}(P)]$ such that $l_i = i$, for $j = 1, \ldots, M$, let $p_{i_j} = i - j/(M + 1)$ let $l_{i_j} = p_{i_j}$. For $j = 1, \ldots, M$, let $r_{i_j} = p_i$. Similarly, for all $i$ such that $r_i = i$, for $j = 1, \ldots, M$, let $p_{i_j} = i + j/(M + 1)$ and let $r_{i_j} = p_{i_j}$. For $j = 1, \ldots, M$, let $l_{i_j} = p_i$.

Now we analyze the size of the platform over time depending on the window chosen by the platform. First, suppose the platform chooses a window that only includes users $i$ in the largest compatible community such that $l_i = p_i$. Then the most extreme user $\min_i p_{i_{\min}}$ could join, and no other users would ever join. The symmetric argument shows that if the window only includes users $i$ in the largest compatible community such that $r_i = p_i$, only one user will ever join the platform.

Now suppose the window includes at least one user with their speech point equal to the left endpoint of their interval and one with their speech point equal to the right endpoint of their interval. Let $I = [L, R]$ be the window set by the platform and let $i^{(L)}$ ($i^{(R)}$) be the index of the user in the largest compatible community with the minimum (maximum) speech point.

Suppose $L \leq i^{(L)} - 1/(M + 1)$ and $R \geq i^{(R)} + 1/(M + 1)$. (That is, suppose the window includes at least one spoiler associated with the most extreme users in the largest compatible community.) Then we claim that there exists a switching order in which no more than 2 users are ever on the platform. The intuition for the choice of switching order is that every user that joins has an interval that covers the one currently on the platform but has a speech point outside of the interval of the user currently on the platform. Thus, if the user that was currently on the platform when this new user joins is asked to leave, they will do so. This swapping process happens forever, so that there are never more than 2 users on the platform.

Formally, for each $i \in s_{\text{opt}}(P)$ inside the window, the following sequence of switches will occur: First, user $i$ will be offered the opportunity to join the platform. Next, if there is any other user on the platform, they will be offered the chance to leave. Next, user $i_1$ will be offered the chance to join, then user $i$ then user $i_2$ then user $i_1$ and so on such that user $i_j$ is offered the chance to join and then $i_{j-1}$ is offered the chance to leave. We claim that at the end of such an iteration, only user $i_M$ (or whatever the maximum $j$ such that $i_j$ is allowed inside the platform) remains. Notice that if user $i$ joins the platform and there is some other user on the platform that $i$ is not compatible with, that user will leave the platform. Also notice that if user $i$ is on the platform and is the only user on the platform user $i_1$ will join and then user $i$ will leave when given the chance. Similarly, user $i_j$ will join the platform if user $i_{j-1}$ is the only one currently on the platform and then user $i_{j-1}$ will leave when given the chance if there is some other user with speech point outside their interval on the platform. Thus, at the end of an iteration $i$, at most one user is left on the platform, and that user is not one in the LCC by the assumption that each user in the LCC has at least one corresponding spoiler allowed on the platform by the window. But since any user remaining on the platform at the end of an iteration is not in the LCC, their only compatibility is with user $i$, which is not on the platform. Thus, in the next iteration when the user on the platform from the previous iteration is allowed to leave the platform, they will do so. Cycling through values of $i$ infinitely in a starvation-free way will give a switching order that is starvation-free for all users, since each spoiler corresponding to a user $i$ is offered the chance to join/stay/leave each time user $i$ is selected for an iteration.

Now consider the situation in which $L > i^{(L)} - 1/(M+1)$. That is, consider the situation in which the left endpoint of the moderation window excludes all spoilers corresponding to user $i^{(L)}$ (The symmetric argument will prove the case when $R < i^{(R)} - 1/(M+1)$, so this is without loss of generality.) We will again show that there exists a switching order that results in $s_{\text{opt}}(P)$ no larger than a constant.

There are three cases to consider. In the first case, suppose that there are greater than $(1-\theta)/\theta$ spoilers corresponding to user $i^{(R)}$. Consider the switching order in which user $i^{(L)}$ joins first. Then, for each $i \in s_{\text{opt}}(P), i \neq i^{(L)}$ inside the moderation window, consider the case in which the switching order is $i, i_1, \ldots, i_{\max}, i, i_1, \ldots, i_{\max}$ (where $i_{\max}$ is the spoiler with largest $j$ inside the moderation window). Notice that $i_{\max} = i_M$ for all $i \neq i^{(R)}, i^{(L)}$ (and possibly for $i = i^{(R)}$ and $i = i^{(L)}$ as well). Notice that if users $i$ and $i^{(L)}$ are the only two users on the platform, user $i_1$ will join the platform when given the chance, then user $i_2$ will as well, leading to all users $i_1, \ldots, i_{\max}$ to join the platform. But then, if all users $i^{(L)}, i, i_1, \ldots, i_{\max}$ are on the platform, user $i$ will have net-negative utility since $\left|\{i_1, \ldots, i_{\max}\}\right| > (1-\theta)/\theta$ by assumption of the window chosen by the platform. Thus, if all users $i^{(L)}, i, i_1, \ldots, i_{\max}$ are on the platform when user $i$ is next offered the chance to leave, they will do so. But then user $i_1$ will have no compatibilities and will all leave when given the chance. Similarly, users $i_2, \ldots, i_{\max}$ will all leave the platform sequentially since they will also have no compatibilities. Thus, any sequence that iterates over values of $i$ repeating such a sequence will result in a platform of size never

more than $2 + M$ which is $O(1)$ since $M$ does not depend on $s_{\text{opt}}(P)$. To make the sequence starvation-free, the sequence should iterate over all $i$ inside the window, and occasionally ask user $i^{(L)}$ whether they want to stay when the platform is otherwise empty, at which point they will have zero utility and stay on the platform.

In the second case, suppose that there are $k < (1-\theta)/\theta - 1$ spoilers corresponding to user $i^{(R)}$. We will use a switching order that is similar to the one for the first case. Let $i^{(L)}$ join the platform first. As before, suppose the switching order is specified by an infinite starvation-free sequence of iterations parametrized by $i \in s_{\text{opt}}(P)$. For a given iteration, suppose the order is $i, i_1, \ldots, i_{\max}, i, i_1, \ldots, i_{\max}$ where again $i_{\max}$ is the user $i_j$ the user with largest $j$ with speech point inside the moderation window. Now suppose that $i = i^{(R)}$ and that $i^{(L)}$ is not compatible with $i^{(R)}$. If $i^{(L)}$ is the only user on the platform when user $i^{(R)}$ is offered the chance to join, user $i^{(R)}$ will not join, and neither will any of the spoilers $i_1^{(R)}, \ldots, i_{\max}^{(R)}$. Now suppose that $i \neq i^{(R)}$ and that $i^{(L)}$ is the only user on the platform. Then, if user $i^{(L)}$ is compatible with user $i$, user $i$ will join the platform as will all spoilers. Then, user $i$ will be given a chance to leave the platform and will do so since they have negative utility as a result of the spoilers. Similarly, all $i_1, \ldots, i_{\max}$ will also cascade off the platform, leaving only user $i^{(L)}$ left on the platform. This proves the case when $i^{(L)}$ is not compatible with $i^{(R)}$, since we just need the iterations to have $i^{(R)}$ go first after $i^{(L)}$ joins, followed by the rest of the iterations over $i \in [s_{\text{opt}}(P)]$ inside the window. (We just need to make sure $i^{(L)}$ is offered a chance to stay when they will do so, say right after user $i^{(L)} + 1$ joins; thus the switching order can be starvation-free.)

Now suppose that $i^{(L)}$ is compatible with $i^{(R)}$. When $i^{(R)}$ is offered the chance to join when $i^{(L)}$ is on the platform, user $i^{(R)}$ will join, as will $i_1^{(R)}, \ldots, i_{\max}^{(R)}$. But since there are fewer than $(1-\theta)/\theta - 1$ spoilers corresponding to user $i^{(R)}$, user $i^{(R)}$ will have net positive utility the second time they are asked and stay. Similarly, all users $i_1^{(R)}, \ldots, i_{\max}^{(R)}$ will also stay. Thus, at the end of the iteration where $i = i^{(R)}$, all users $i^{(L)}, i^{(R)}, i_1^{(R)}, \ldots, i_{\max}^{(R)}$ will be present on the platform.

Now, consider some other iteration $i$ for $i \neq i^{(R)}$ but after $i^{(R)}$ and all corresponding spoilers have joined the platform. The fact that $i^{(R)}$ and $i^{(L)}$ are mutually compatible (by assumption) implies that for any $i$ such that $i^{(R)} < i < i^{(L)}$ either $i^{(R)}$ or $i^{(L)}$ is compatible with user $i$. But since there are a total of fewer than $(1-\theta)/\theta$ users on the platform and user $i$ is compatible with at least one of them, user $i$ will join the platform when given the chance. User $i_1$ will also join the platform since they have at least one compatibility (user $i$) and there are no more than $(1-\theta)/\theta$ users on the platform. Similarly, users $i_2, \ldots, i_M$ will also join the platform since they will also have no fewer than a $\theta$ fraction of compatibilities. Next, user $i$ will be given an opportunity to leave, and now there are $3 + k + M$ users on the platform (users $i, i^{(R)}, i^{(L)}, k$ spoilers corresponding to user $i^{(R)}$ and $M$ spoilers corresponding to user $i$) and user $i$ has at most $1 + k$ compatibilities. Thus, they have negative utility and will leave. At this point, user $i_1$ will have zero compatibilities and will also leave, then user $i_2$ and the rest of the spoilers corresponding to user $i$ will cascade off the platform. Thus, any

permutation over $\{i : i^{(L)} < i < i^{(R)}\}$ where user $i^{(R)}$ is offered the chance to stay periodically and the sequence $i^{(L)}, i_1^{(L)}, \ldots, i_{\max}^{(L)}$ is interspersed periodically in between iterations. Thus, there exists a switching order where, if $i^{(L)}$ and $i^{(R)}$ are mutually compatible, at most a finite number of users are on the platform at any one time. This completes the construction of a switching order for the second case.

In the third case, suppose that there are $k \in [(1 - \theta)/\theta - 1, (1 - \theta)/\theta]$ spoilers corresponding to user $i^{(R)}$ inside the window. Our strategy will be similar to the second case: We will allow user $i^{(L)}$ on the platform, then user $i^{(R)}$ then the first $j = 1, \ldots, \lfloor (1 - \theta)/\theta - 1 \rfloor$ spoilers associated with $i^{(R)}$ on the platform, then sequences of $i, i_1, \ldots, i_M, i, i_1, \ldots, i_M$ for each $i$ such that $i^{(L)} < i < i^{(R)}$. The only hitch is there are more than $(1 - \theta)/\theta - 1$ spoilers associated with user $i^{(R)}$, and to have a starvation-free switching order, these users must have the chance to join the platform infinitely often in the switching order. So we need to consider whether there exist switching orders where these additional users can be asked to join where they will not do so. We can notice that there are indeed such switching order if we use the switching order from case 2 and then strategically insert the last $\lfloor (1 - \theta)/\theta - 1 \rfloor - k$ spoilers corresponding to user $i^{(R)}$ into the order at a position where these users are not willing to join the platform. In fact such a position exists at any time when a user $i$ and all spoilers $i_1, \ldots, i_M$ have joined the platform for $i^{(L)} < i < i^{(R)}$ but before they all cascade off the platform. This can be done in a starvation-free way and completes this case.

Next, we deal with the condition when $\theta \in (1/3, 1/2]$. The construction is similar to $\theta \le 1/3$, but there is some additional analytical complexity to manage. The chief reason for the complexity is that, since $\theta$ is larger than $1/3$, it may be the case that users in the LCC would join the platform when given the chance but their corresponding spoilers would not join the platform. Our construction will ameliorate this issue and ensure that there is a switching order in which, whenever a user in the LCC joins the platform, so will their corresponding spoilers.

The construction is as follows. We will differentiate two different types of spoilers: *long spoilers* and *short spoilers*. Long spoilers will be the first few spoilers (the ones with the first indices $j$ for spoiler $i_j$) and short spoilers will be the remaining spoilers. Long spoiler $i_j$ will have an interval that extends beyond the speech of user $i$ to cover a few more users. Short spoiler $i_j$ will have an interval that extends just until the speech of user $i$. Long spoiler $i_j$ will cover enough users so that they will join the platform whenever user $i$ joins, for some switching order. Short spoiler $i_j$ will also join whenever user $i$ and all long spoilers corresponding to user $i$ have joined for the switching order we specify.

Formally, suppose there is some user $i$ such that $1 \le i < \lceil \theta s_{\text{opt}}(P) \rceil$ or $\lceil \theta s_{\text{opt}}(P) \rceil + 1 < i \le s_{\text{opt}}(P)$. This user will have $6 + 3\lfloor 1/\theta \rfloor$ corresponding long spoilers. If $0 \le i < \lceil \theta s_{\text{opt}}(P) \rceil$, the long spoiler $i_j$ will have $l_{i_j} := p_{i_j}$ and

$$r_{i_j} := i + \frac{1}{M+1}(3 + \lfloor 1/\theta \rfloor).$$

If $\lceil \theta s_{\text{opt}}(P) \rceil + 1 < i < s_{\text{opt}}(P)$, the spoiler $i_j$ will have $r_{i_j} := p_{i_j}$ and

$$l_{i_j} := i - \frac{1}{M+1}(3 + \lfloor 1/\theta \rfloor).$$

For users $i = \lceil \theta s_{\text{opt}}(P) \rceil, \lceil \theta s_{\text{opt}}(P) \rceil + 1$, there will be

$$\left\lceil \frac{\theta}{1 - \theta}(5 + \lfloor 1/\theta \rfloor) \right\rceil - 1$$

corresponding long spoilers. If user $i = \lceil \theta s_{\text{opt}}(P) \rceil - 1$, the long spoiler $i_j$ will have $l_{i_j} := p_{i_j}$ and

$$r_{i_j} = i + \frac{2}{M+1}.$$

If user $i = \lceil \theta s_{\text{opt}}(P) \rceil$, the long spoiler $i_j$ will have $r_{i_j} := p_{i_j}$ and

$$l_{i_j} = i - \frac{2}{M+1}.$$

Each user will have a total of

$$M := \max \left\{ \left\lfloor \frac{2}{\theta}(2 + \lfloor 1/\theta \rfloor) \right\rfloor, \right.$$
$$6 + 3\lfloor 1/\theta \rfloor,$$
$$\left. \left\lfloor \frac{3 + \lfloor 1/\theta \rfloor}{\theta} \right\rfloor + 5 + 3\lfloor 1/\theta \rfloor \right\}$$

corresponding spoilers (the number of short spoilers can be calculated by subtracting away the number of long spoilers).

As before, we will break the analysis for $\theta \le 1/3$ into several cases. First, consider when the platform chooses a window that only includes users $i$ in the largest compatible community such that $l_i = p_i$. Then the most extreme-speech user $\min_i p_{i_{\min}}$ could join the platform and no other user would be willing to join. The symmetric argument shows that if the window only includes users $i$ in the largest compatible community such that $r_i = p_i$ only one user will ever join the platform if the most extreme-speech user joins first.

Now, consider when the platform chooses a window with at least one user $i$ with $l_i = p_i$ and $r_i = p_i$ for $i \in [s_{\text{opt}}(P)]$. As before, $I = [L, R]$ and let $i^{(L)}$ ($i^{(R)}$) be the index of the user in the largest compatible community with the minimum (maximum) speech point.

Suppose $L \le i^{(L)} + 1/(M+1)$ and $R \ge i^{(R)} + 1/(M+1)$. (That is, suppose the window includes at least one spoiler corresponding to the most extreme users in the largest community community.) If $i^{(L)} = \lceil \theta s_{\text{opt}}(P) \rceil - 1$ and $i^{(R)} = \lceil \theta s_{\text{opt}}(P) \rceil$, there are only a constant number of users allowed on the platform, so the bound holds, regardless of switching order. If, without loss of generality, $i^{(L)} = \lceil \theta s_{\text{opt}}(P) \rceil - 1$ but $i^{(R)} \ne \lceil \theta s_{\text{opt}}(P) \rceil$, if $i^{(R)}$ is compatible with the long spoilers corresponding to user $i^{(L)}$, there are only a constant number of users allowed on the platform, so the bound holds, regardless of switching order. If, without loss of generality, $i^{(L)} = \lceil \theta s_{\text{opt}}(P) \rceil - 1$ and $i^{(R)} \ne \lceil \theta s_{\text{opt}}(P) \rceil$ and $i^{(R)}$ is not compatible with the long spoilers corresponding to user $i^{(L)}$, then we can use the same switching order that we use in the case that $i^{(L)} \ne \lceil \theta s_{\text{opt}}(P) \rceil - 1$ and $i^{(R)} \ne \lceil \theta s_{\text{opt}}(P) \rceil$, the description of which follows next.

If $i^{(L)} \ne \lceil \theta s_{\text{opt}}(P) \rceil - 1$ and $i^{(R)} \ne \lceil \theta s_{\text{opt}}(P) \rceil$, the following switching order yields $s(P, I) = O(1)$. As in the case of $\theta < 1/3$,

$L \leq i^{(L)} - 1/(M+1)$ and $R \geq i^{(R)} + 1/(M+1)$, we claim that for each user, there is some user who is compatible with them that they are not compatible with. We can choose a switching order such that each time a user joins the platform, someone who is compatible with them but who they are not compatible with will join. Then, if the original user is asked if they want to leave, they will do so and the process repeats. The fact that for each user, there is some user who is compatible with them that they are not compatible with can be verified by the fact that each spoiler is not compatible with the user in the LCC that is furthest away from them in the window, but this LCC user is compatible with them. This concludes the analysis for when $L \leq i^{(L)} + 1/(M+1)$ and $R \geq i^{(R)} + 1/(M+1)$.

Now, consider when $L > i^{(L)} - 1/(M+1)$. That is, consider the situation in which the left endpoint of the moderation window excludes all spoilers corresponding to user $i^{(L)}$. (The symmetric argument proves the case when $R < i^{(R)} + 1/(M+1)$.) Let $k$ be the number of spoilers corresponding to user $i^{(R)}$. This time, there are only the two cases: the first case is the one where $k > (1-\theta)/\theta$ and the second case is when there are $k \leq (1-\theta)/\theta$ spoilers. (There is no need for the case when $k < (1-\theta)/\theta - 1$ since this number is less than 1 for $\theta > 1/3$.)

For the first case, we consider when $k > (1-\theta)/\theta$. This case is simple: we can use the switching order we used when $\theta \leq 1/3$ and $k > (1-\theta)/\theta$. For the switching order leading to small $s(P, I)$ in that case, we can use the same switching order and it will lead to a small $s(P, I)$ in this case as well. As before, we choose a switching order such that $i^{(L)}$ always chooses the stay on the platform, and then a starvation free switching order composed of concatenated sequences $i, i_1, \ldots, i_{\max}, i, i_1, \ldots, i_{\max}$ for each $i \in [s_{opt}(P)]$ leads one of two possibilities to occur: either none of the users join the platform, or they all do during the first part of the sequence $(i, i_1, \ldots, i_{\max})$ and then leave during the remaining part of the sequence.

For the second case, we consider when $k \leq (1-\theta)/\theta$. This case, we will choose the switching order carefully so that, when a user $i$ in the LCC joins the platform, so do their spoilers, leading to a cascade of users $i, i_1, \ldots, i_M$ all leaving the platform. This case is the reason why the construction for $\theta \leq 1/3$ would not work for when $\theta \in (1/3, 1/2)$. For our switching order, we will first let user $i^{(L)}$ join the platform, followed by user $i^{(R)}, i_1^{(R)}, \ldots, i_{\max}^{(R)}$. Then, exactly 2 users from the LCC with the smallest indices $i$ such that $i < \lceil \theta s_{opt}(P) \rceil$ will be next in the switching order and will join the platform. Next, user $\lceil \theta s_{opt}(P) \rceil$ will be next in the switching order and join the platform, followed by all corresponding spoilers. By definition of their intervals, each of these spoilers will join the platform. Then, user $\lceil \theta s_{opt}(P) \rceil - 1$ will have a chance to leave the platform and will do so. Similarly, the first $M - (3 + \lfloor 1/\theta \rfloor)$ corresponding spoilers will have a chance to leave and will do so. Then, user $\lceil \theta s_{opt}(P) \rceil - 2$ will have a chance to join the platform and will do so, as will all corresponding spoilers Then, the remaining $3 + \lfloor 1/\theta \rfloor$ spoilers corresponding to user $\lceil \theta s_{opt}(P) \rceil - 1$ will have a chance to leave the platform and will do so. Then the first $M - (3 + \lfloor 1/\theta \rfloor)$ spoilers corresponding to user $\lceil \theta s_{opt}(P) \rceil - 2$ will have a chance to leave the platform and will do so. The same pattern will repeat for each $i \in [s_{opt}(P)]$ such that $i < \lceil \theta s_{opt}(P) \rceil - 1$, where the switching order for the $i$th iteration will be $i, i_1, \ldots, i_M, i, i_1, \ldots, i_M$. In each

case, users $i, i_1, \ldots, i_M$ will join the platform and then and remaining spoilers corresponding to user $i + 1$ will have the chance to leave the platform and do so. After all $i^{(L)} < i < \lceil \theta s_{opt}(P) \rceil - 1$ iterations have been completed, the mirror process will occur: First, exactly 2 users from the LCC will join indexed by the largest indices less than or equal to $\lceil \theta s_{opt}(P) \rceil - 1$. Subsequently, user $\lceil \theta s_{opt}(P) \rceil$ will join the platform, then will all corresponding spoilers. Then, the first $M - (3 + \lfloor 1/\theta \rfloor)$ spoilers corresponding to user $\lceil \theta s_{opt}(P) \rceil$ will have a chance to leave the platform and will do so. Then, for $i = \lceil \theta s_{opt}(P) \rceil, \ldots, i^{(R)}$, the switching order will be $i, i_1, \ldots, i_{\max}$, followed by any remaining spoilers corresponding to user $i - 1$, then the first $M - (3 + \lfloor 1/\theta \rfloor)$ spoilers corresponding to user $i$ will have a chance to leave the platform and will do so. The numbers of spoilers that remain in each iteration are set so that spoilers in the next iteration are willing to join the platform, at which point all of those remaining spoilers leave, triggering a cascade of users who leave the platform. This completes the case when $k \leq (1-\theta)/\theta$. $\square$

**Theorem 3.3.** *For any population $P$ such that $\theta_i > 1/2$ for all $i$, it holds that, for any $\beta \in (0, 1)$, given a random sample of $m$ users, a platform can find a moderation window $I \in \mathcal{I}$ in time polynomial in $m$ such that*

$$\Pr\left[s(P, I) \geq \beta \cdot (2\theta_{\min} - 1)s_{opt}(P)\right]$$

$$\geq 1 - n \exp\left\{-m\left(\left(\theta_{\min} - \frac{1}{2}\right)\frac{s_{opt}(P)}{n}(1-\beta)\right)^2\right\},$$

*where the probability is over the sample of $m$ users. In fact, sampling $m$ users uniformly at random and applying the window achieving the lower bound in Theorem 3.2 on the sample to the full population will achieve the stated bound.*

**Proof of Theorem 3.3.** We first define some random variables that we will use in the rest of the proof. Let $\mathcal{S}$ be a sample of users of size $m$. Let $\mathcal{U}$ to be the largest set of users in the sample such that each user in the set is mutually compatible with every other user in the set. Let $u := |\mathcal{U}|$. We proved in algorithm 2 that this set can be found in $O(m^3)$ time. For a fixed interval $[a, b]$, let $\mathcal{S}_{a,b} := \{i \in \mathcal{S} : a \leq p_i \leq b\}$. Define

$$\mathcal{U}_{a,b} = \{i \in [n] : a \leq p_i \leq b, l_i \leq a, b \leq r_i\}$$

to be the set of users whose speech falls in $[a, b]$ and intervals cover $[a, b]$. Let $u_{a,b} = |\mathcal{U}_{a,b}|$. Define $\mathcal{V} = \mathcal{U} \cap \mathcal{S}$, and let $V = |\mathcal{V}|$. Define $\mathcal{V}_{a,b} = \mathcal{U}_{a,b} \cap \mathcal{S}_{a,b}$, and let $V_{a,b} = |\mathcal{V}_{a,b}|$.

Notice

$$\mathbb{E}[V_{a,b}] = \sum_{i \in \mathcal{U}_{a,b}} \mathbb{E}[\mathbb{1}\{i \in \mathcal{S}\}]$$

$$= \sum_{i \in \mathcal{U}_{a,b}} \frac{m}{n}$$

$$= \frac{m}{n}u_{a,b}.$$

For some $t$, define the event

$$\mathscr{A}_{a,b} = \left\{\left|V_{a,b} - \frac{m}{n}u_{a,b}\right| \leq t\right\}.$$

We will upper bound the probability of $\mathscr{A}_{a,b}$ as a function of $t$. Let $X_1, \ldots, X_m$ represent the indicator variable that is 1 if the

corresponding element in $\mathcal{S}$ is in $\mathcal{U}_{a,b}$ and zero otherwise. To do this, we will use Hoeffding's inequality:

**Lemma G.3** (Hoeffding's inequality for sampling without replacement, adapted from [5]). *Let a population consist of $n$ values $x_1, x_2, \ldots x_n$ where $x_i \in [a, b]$ for all $i \in [n]$, and let $X_1, X_2, \ldots, X_k$ be a sample of $k$ values from the population without replacement. Define $S = X_1 + \cdots + X_k$. Then*

$$\mathbb{P}\left[\left|S - \mathbb{E}[S]\right| \geq t\right] \leq 2\exp\left\{\frac{-2t^2}{k(b-a)^2}\right\}.$$

The desired inequality can be derived as:

$$\mathbb{P}\left[\overline{\mathscr{A}_{a,b}}\right] = \mathbb{P}\left[\left|V_{a,b} - \frac{m}{n}u_{a,b}\right| \geq t\right]$$

$$= \mathbb{P}\left[\left|V_{a,b} - \mathbb{E}[V_{a,b}]\right| \geq t\right]$$

$$= \mathbb{P}\left[\left|\sum_{i=1}^{m} X_i - \mathbb{E}\left[\sum_{i=1}^{m} X_i\right]\right| \geq t\right]$$

$$\leq 2\exp\left\{\frac{-2t^2}{m}\right\}. \quad \text{(Hoeffding's inequality, Lemma G.3)}$$

Let $\mathscr{A}$ be the event that $\mathscr{A}_{p_i, p_j}$ holds for all $i < j \in [n]$. Then, using a union bound, we have

$$\mathbb{P}\left[\overline{\mathscr{A}}\right] \leq 2n^2 \exp\left\{\frac{-2t^2}{m}\right\}$$

If we would like $\mathscr{A}$ to occur with probability at least $1 - \delta$, we can choose

$$t = \sqrt{\frac{m}{2}\log\left(\frac{2n^2}{\delta}\right)}.$$

Define the random variables

$$L := \min_{j \in \mathcal{V}}\{p_j\},$$

$$R := \max_{j \in \mathcal{V}}\{p_j\}.$$

to represent the least and greatest speech points in $\mathcal{V}$. Notice that by definition $\mathcal{V}_{L,R} = \mathcal{V}$.

Conditioning on $\mathscr{A}$, we have that

$$u_{L,R} \geq \frac{n}{m}\left(V_{L,R} - \sqrt{\frac{m}{2}\log\left(\frac{2n^2}{\delta}\right)}\right).$$

Now we we use $\mathscr{A}$ to prove a lower bound on the size of $V_{L,R}$. Then, $\mathscr{A}$ also implies

$$V \geq \frac{m}{n}u - \sqrt{\frac{m}{2}\log\left(\frac{2n^2}{\delta}\right)}$$

$$\geq \frac{m}{n}(2\theta - 1)s_{\text{opt}}(P) - \sqrt{\frac{m}{2}\log\left(\frac{2n^2}{\delta}\right)}.$$

where the second inequality comes from Lemma G.2 that $u \geq (2\theta - 1)s_{\text{opt}}(P)$.

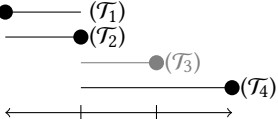

**Figure 10: Construction for a population instance where better personalization reduces the size of the platform achievable with a moderation window. Users in $\mathcal{S}_0$ are shown in black.**

Putting it together, conditionine on $\mathscr{A}$ and setting the window to $[L, R]$, it holds that

$$s(P, [L, R]) = u_{L,R}$$

$$\geq \frac{n}{m}\left(V_{L,R} - \sqrt{\frac{m}{2}\log\left(\frac{2n^2}{\delta}\right)}\right)$$

$$\geq \frac{n}{m}\left(V - \sqrt{m\log\left(\frac{n}{\delta}\right)}\right) \quad \text{(assuming } \delta \leq 1/2)$$

$$\geq \frac{n}{m}\left(\frac{m}{n}u - 2\sqrt{m\log\left(\frac{n}{\delta}\right)}\right)$$

$$\geq (2\theta - 1)s_{\text{opt}}(P) - 2n\sqrt{\frac{1}{m}\log\left(\frac{n}{\delta}\right)},$$

which can be rearranged to show the desired result.

## G.2 Proofs for Section 5

**Proposition 5.2.** *There exist two populations $P := \{l_i, p_i, r_i, b_i, \lambda_i\}, \mathcal{S}_0$, $P' := \{l_i, p_i, r_i, b_i, \lambda_i'\}, \mathcal{S}_0$ that differ only by the fact that personalization in the second is strictly better for every user than in the first (i.e, $\lambda_i > \lambda_i'$ for all $i \in [n]$) such that*

$$s(P, I^*) > s(P', I^{*'}) \tag{1}$$

*for $I^*$, $I^{*'}$ optimal choices of window-based policies on $P$ and $P'$ respectively. In fact, such a pair of populations satisfying ineq. (1) can be constructed for any $b := b_1 = \cdots = b_n$ and $\lambda := \lambda_1 = \cdots = \lambda_n$ and an appropriately chosen $\lambda' := \lambda_1' = \cdots = \lambda_n'$.*

**Proof of Proposition 5.2.** We will build a family of problem instances where the best window for a platform with worse personalization results in a larger platform than the best window for a platform with better personalization.

Our construction is as follows. For concreteness, we will instantiate the construction with numbers, but of course, the family is general. We create 4 types of users indexed by $\mathcal{T}_1, \ldots, \mathcal{T}_4$, where the size of each set will be determined later. The speech point of user $i \in \mathcal{T}_j$ will be $p_i = j$. All users in $\mathcal{T}_1, \mathcal{T}_2$ will have interval $[1, 2]$, users in $\mathcal{T}_3$ will have interval $[2, 3]$ and users in $\mathcal{T}_4$ will have interval $[2, 4]$. Users in $\mathcal{T}_1, \mathcal{T}_2$ and $\mathcal{T}_4$ start on the platform: $\mathcal{S}_0 = \mathcal{T}_1 \cup \mathcal{T}_2 \cup \mathcal{T}_4$. A drawing of the construction is depicted in fig. 10.

Let $\theta := \lambda b/(1 + \lambda b)$ and $\theta' := \lambda'b/(1 + \lambda'b)$. Define $\beta := \theta'/\theta$ and notice $\beta \in (0, 1)$ by the fact that there is more personalization

                                                                       

for $P'$ than for $P$. The construction only works if

$$\beta \geq \frac{\lceil \theta n \rceil + 1}{n - 1};$$

$\lambda'$ must not be too much smaller than $\lambda$.

Now we define the sizes of each of the sets of users. Define $n_0 := |\mathcal{S}_0| = |\mathcal{T}_1| + |\mathcal{T}_2| + |\mathcal{T}_4|$. We will have

$$|\mathcal{T}_1| = \lceil \theta(n_0 - 1) \rceil + 1 - |\mathcal{T}_2|;$$
(3)

$$\max \left\{ (2\theta - 1)n_0 + 1, \beta\theta(n_0 - 1) \right\} < |\mathcal{T}_2| < \theta(n_0 - 1);$$
(4)

$$\frac{(1 - \beta)\lceil \theta n_0 \rceil + 1}{\beta\theta} < |\mathcal{T}_3| < \lfloor (1 - \theta)(n_0 - 1) \rfloor;$$
(5)

$$|\mathcal{T}_4| = \lfloor (1 - \theta)(n_0 - 1) \rfloor.$$
(6)
(7)

First, notice that $\mathcal{S}_0$ is a compatible set. This is because all users in $\mathcal{T}_1, \mathcal{T}_2$ and $\mathcal{T}_4$ have $\lceil \theta(n_0 - 1) \rceil$ compatibilities, which is a $\theta$ fraction of the total size of $\mathcal{S}_0$. Also, notice that no users in $\mathcal{T}_3$ will join the platform, since they have strictly fewer than the $\theta(n_0 - 1)$ compatibilities necessary for users in $\mathcal{T}_3$ to join.

However, under more personalization, users in $\mathcal{T}_3$ *would* be willing to join the platform if they were inside the moderation window, since they have at least $\beta\theta(n_0 - 1)$ compatibilities. Further, under switching orders where, after users in $\mathcal{T}_3$ are offered to join the platform and then users in $\mathcal{T}_1$ and $\mathcal{T}_2$ were given the chance to leave, they would all do so since they would not have the $\beta\theta(n - 1)$ compatibilities necessary to be willing to stay. These users would stay off the platform permanently, since they would never have

enough compatibilities going forward. Thus, the platform would be smaller than $n_0$ after switching stabilized. This shows us that, if the window were set to encompass all users, more personalization is not better: the platform could end up smaller than it would under less personalization. Similarly, the platform could not set a window that captured $n_0$ under more personalization. This proves the desired result.

## G.3 Proofs for Section 6

**Proposition B.1** (Corollary to Proposition 5.2). *There exist users $\{l_i, p_i, r_i, b_i\}_{i \in [n]}$ and two platforms Platform 1 and Platform 2 where:*

(1) *personalization for Platform 1 is no worse than on Platform 2 for every user (i.e., $\lambda_{i,1} \leq \lambda_{i,2}$ for all $i \in [n]$),*

(2) *all individuals in a largest compatible community start on Platform 1 and Platform 2 is empty,*

*and at equilibrium, Platform 2 can set a window so at least an $\min\left\{1/(1 + b), b/(1 + b)\right\}$ fraction of users in the largest compatible community are on Platform 2. In fact, this result holds for any fixed choices of $b := b_1 = \cdots = b_n$ and $\lambda := \lambda_1 = \cdots = \lambda_n$ and for some appropriately chosen $\lambda' := \lambda'_1 = \cdots = \lambda'_n$.*

**Proof of Proposition B.1.** We use the same construction used in the proof of Proposition 5.2. In Proposition 5.2, we proved that a platform with more personalization was not necessarily better off than one with less personalization. In fact, we showed that the platform could capture at most a $\max\left\{n - |\mathcal{T}_4|, |\mathcal{T}_4|\right\}$ fraction of the platform. Platform 2 can set a window to capture either $\mathcal{T}_1, \mathcal{T}_2$ or $\mathcal{T}_4$, depending on which one is smaller, resulting in a proportion of $\min\{1/(1 + b), b/(1 + b)\}$.

