# OpenReview forum: "Content Moderation and the Formation of Online Communities: A Theoretical Framework"
_ACM.org/TheWebConf/2024/Conference — TheWebConf24 Oral_

### Official Review · Reviewer_MqFq · 2023-11-19

**Novelty:** 3
**Technical Quality:** 3

**Review:**

This paper presents a framework to analyze content moderation, user participation decisions and their impacts on communities.

Strengths:
The paper investigates an important problem of content moderation policies, and characterize the effectiveness of moderation policies for creating and sustaining stable online communities.

Weaknesses:
Related work should be an independent section.

This paper has the length of 24 two-column pages (including appendix). It is hard to read the paper within 8 pages without reading the appendix. So this paper might be more suitable for submitting to a journal rather than a conference.

A summarization of findings or take-away lessons what we can learn from the analysis is missing in this paper.

**Questions:**

Weaknesses:
Related work should be an independent section.

This paper has the length of 24 two-column pages (including appendix). It is hard to read the paper within 8 pages without reading the appendix. So this paper might be more suitable for submitting to a journal rather than a conference.

A summarization of findings or take-away lessons what we can learn from the analysis is missing in this paper.

**Reviewer Confidence:**

3: The reviewer is confident but not certain that the evaluation is correct

**Scope:**

3: The work is somewhat relevant to the Web and to the track, and is of narrow interest to a sub-community

---

### Official Review · Reviewer_DiNa · 2023-11-24

**Novelty:** 5
**Technical Quality:** 4

**Review:**

Summary: The paper presents a mathematical model for the effect of social media content moderation decisions on users' choices to join or remain on a platform. It models content on a linear scale, users' and platforms' "tolerance" for content as an interval, and users' content production as a point on the scale.

Reasons to accept:
* a timely, important topic with clear relevance to the Web
* if the model is a reflection of reality, it can help platforms evaluate and set content moderation policies
* the paper is very well written and accessible even to readers who do not (seek to) understand the model in detail

Reasons to reject:
* it is unclear whether the model reflects reality accurately enough
* the paper does not strike a good balance between what is in the main part of the paper and the very long appendix; the main part of the paper alone (without the appendix) does not permit a thorough evaluation of the claims made by the authors

I would like to start by congratulating the authors for a very well written paper. The accessible writing provides a good intuition for the model and takeaways even for readers who are not experts. The topic is an important and timely one, and giving social media platforms a means to gauge the impact of their content moderation policy is a clear practical benefit.

That being said, it is unclear how realistic the assumptions underlying the model are. The findings intuitively make sense, but content moderation in practice can diverge from the model in important ways. Firstly, the model assumes that content moderation is perfect, but in practice it is not, in that platforms enforce their rules inconsistently, and the criticism against platforms often arises from such cases. Platforms are known to grant policy exemptions to participants they deem important, and they make mistakes when moderating content (false negatives and false positives). Prior research has shown that some demographics or types of content have a higher likelihood of false positives/negatives (https://dl.acm.org/doi/pdf/10.1145/3479610), for example when judgement calls need to be made that automated methods cannot handle well, and that overworked human content moderators can easily get wrong. Furthermore, content moderation is not always immediate; when platforms review content manually, it is usually permitted to remain active while the review is underway, and processing delays can be relatively long compared to the short engagement lifetime of content on social media, i.e. many users may have been exposed to content that is ultimately removed by the platform. (In these cases, content moderation is only effective for those users who have not yet seen the content, whereas the users who _have_ seen the content before it was deleted may incorrectly believe that the platform tolerates that kind of content.) More generally, due to limited transparency, users have an imperfect picture of the platforms' content moderation, giving rise to folk theories and dissatisfaction even when the policy as written aligns with users' preferences, because the policy as written is not what users perceive to be occurring on the platform. This all makes it seem that a linear, perfect model for content and content moderation may be an oversimplification.

Another issue I would like to flag is the extensive use of the appendix. Various sections of the paper refer the reader to the appendix (Section 6 is an extreme example that only summarizes the appendix without much usefulness on its own). It is challenging to judge the paper based on its main part alone without referring to the appendix. I understand that the main part of the paper cannot hold all the proofs but in this case the authors may have gone overboard with trying to make too many different results fit into the paper.

Minor comments:
* In Figure 2, for clarity the authors could note that user 5 would not tolerate content from users 1+2, but users 1+2 would tolerate user 5.
* I found the personalization example (Section 6) suboptimal because ideally a personalization system would not only pick a fixed number of topics of interest for a given user, but rather can account for different users having a different number of interests (i.e., broad vs. only specific interests).
* Additional related work: https://papers.ssrn.com/sol3/papers.cfm?abstract_id=4307346

**Questions:**

How does imperfect/inconsistent content moderation affect the validity of the results presented in the paper?

**Ethics Review Description:**

no ethical issues

**Reviewer Confidence:**

1: The reviewer's evaluation is an educated guess

**Scope:**

3: The work is somewhat relevant to the Web and to the track, and is of narrow interest to a sub-community

---

### Official Review · Reviewer_qPuT · 2023-11-29

**Novelty:** 4
**Technical Quality:** 4

**Review:**

The paper presents a theoretical model analyzing the impact of content moderation policies on online community participation. It explores how these policies can paradoxically increase user engagement and diversify content, despite their restrictive nature. The authors delve into various aspects, including the effects of moderation on resource-limited platforms, and the role of personalization. This study is timely and relevant, given the current discourse on social media's societal impact. The approach is innovative, offering a mathematical framework for understanding complex social phenomena allowing to simulate situations which are not easy to observe in real world, before content moderation decisions are made.

* Strengths:
    * The model is comprehensive, addressing multiple facets of content moderation, such as user preference alignment, and resource constraints.
    * The paper's strength lies in its depth of analysis, using a metric space to represent content and user preferences. This allows for a nuanced understanding of content moderation's impact on user utility and platform participation
    * The study offers valuable insights for social media platforms and policymakers on the nuanced impacts of content moderation.

* Weaknesses:
    * The model's effectiveness hinges on certain assumptions about user behavior and preferences. These assumptions may not always hold true in real-world settings, potentially limiting the model's applicability. While the model attempts to capture complex social interactions, it may oversimplify the intricacies of real-world social dynamics.
    * The mathematical rigor, while a strength, also poses a challenge for practical application. The model's complexity might hinder its use by practitioners or policymakers.
    * The absence of empirical data to support or validate the model is a notable gap. Incorporating real-world case studies or data could significantly enhance the paper's validity.

Comments:

The paper posits that content moderation policies could simultaneously increase participation and diversify the content available on platforms. However, it's important to note that these two outcomes may not always be exclusive. While increasing participation and diversifying content can both be beneficial, they may not necessarily occur concurrently.

Furthermore, the paper's heavy reliance on appendices for detailed content, notably with all of Section 6 referencing material solely located in the appendix, detracts from its coherence. This approach is not ideal, as it burdens the reader with excessive cross-referencing, leading to a fragmented understanding of the study's full implications. In my review, I did not delve into these appendix sections due to their excessive and somewhat disjointed nature.

Future work could conduct studies using data from actual online platforms to validate the model's predictions and assumptions and expanding the model to cover a broader spectrum of user behaviors and preferences. For instance, since most of reddit data including some of the moderators choices is public, this might be a good source to validate the model. This will really make the paper strong and help in making a case for the external validity of the model.
It would also help to provide a simplified version or guidelines for practitioners to implement the model in real-world scenarios.

**Questions:**

I do not have any clarifications.

**Reviewer Confidence:**

3: The reviewer is confident but not certain that the evaluation is correct

**Scope:**

4: The work is relevant to the Web and to the track, and is of broad interest to the community

---

### Official Review · Reviewer_ir4W · 2023-11-29

**Novelty:** 6
**Technical Quality:** 5

**Review:**

Summary: The paper proposes a theoretical framework to model how content moderation policies under various circumstances affect the user community sizes in online  social media platforms.

Strengths:

1.	Interesting counter-intuitive results backed by theoretical proofs.

2.	Good readability due to the inclusion of real-world examples explaining certain theoretical propositions.

3.	Rigorous and in-depth theoretical investigation of the research problem.


Weaknesses:

1.	Motivation behind certain formulations are not clear and sometimes confusing. (i) Why use a single speech point for user generated content? Users can generate content that may fall on different sides of the spectrum (i.e., multiple posts); (ii) Why the moderation window has to be a single continuous interval? Isn’t it more realistic to represent moderation as a set of intervals (e.g., fringe elements on either side of the spectrum)?

2.	The design setup of allowing users to join or leave platform _one at a time_  is also limiting and perhaps does not reflect real world scenario. Given the scale of social media platforms, multiple users may be independently making decisions to join/leave _in parallel_ by observing the same content in the same time duration, thereby not influencing each other in any way.

3.	No discussion on limitations of the proposed framework.

4.	No empirical evidence to reinforce the theoretical findings.

**Questions:**

Please see Weaknesses in the review (copied down below for convenience). Other than the ones listed there, I have the following question: In section 3.1, a special case of theta = 1 is considered. Looking at the definition of theta (line 386), I am wondering how this is possible?

**Reviewer Confidence:**

3: The reviewer is confident but not certain that the evaluation is correct

**Scope:**

3: The work is somewhat relevant to the Web and to the track, and is of narrow interest to a sub-community

---

### Official Review · Reviewer_JWMr · 2023-12-01

**Novelty:** 7
**Technical Quality:** 6

**Review:**

This paper introduces a novel theoretical model for content moderation and user participation in online communities. The model is grounded in simple yet plausible assumptions, including the concept of a 'speech point' and a 'tolerance interval' for each user within a one-dimensional metric space of potential content. From these assumptions, the paper derives a clear utility function for users, taking into account the speech points of others in the community, the platform's window-based moderation policy, and the level of content personalization of the platform.

The authors then use this utility function to conduct a comprehensive theoretical analysis. This analysis not only illuminates the dynamic interplay between a platform's moderation policy, its personalization level, and overall sustainability, but also leads to several critical and counterintuitive conclusions. The proposed model offers a valuable framework for analyzing, explaining, and optimizing various real-world issues related to content moderation in online communities. The theoretical results are clean and nicely interpreted and linked to practice. The authors have also nicely articulated the implications of their findings to real platforms and the limitations and potential future analyses that can be built upon their framework.  Overall, the paper is a pleasure to read.

While there isn't really much to say about the weakness of the nice piece of work, I notice that the model doesn't consider a critical factor, the capacity of a user to consume content on the platform (or it assumes that a user reads all content that are not censored), and therefore every other user j will contribute to the utility (or disutility) of a user.  Intuitively, adding this realistic constraint might affect some of the conclusions, especially the conclusions about personalization.  I see that the authors has considered the capacity constraints (i.e., the consumption bandwidth in Appendix B.1) when analyzing the competition among platforms.  It'll be nice to comment on whether having this constraint will affect the conclusions presented in Section 3-5, especially those about moderation policy and personalization.

In conclusion, the paper proposes a simple and elegant theoretical model for a critical, widely-concerned problem in online communities, which leads to multiple surprising and practically useful findings.

**Questions:**

See the review above. More discussion about the impact of the "consumption bandwidth" constraint would be appreciated.

**Reviewer Confidence:**

4: The reviewer is certain that the evaluation is correct and very familiar with the relevant literature

**Scope:**

4: The work is relevant to the Web and to the track, and is of broad interest to the community

---

### Official Review · Reviewer_sceY · 2023-12-01

**Novelty:** 3
**Technical Quality:** 3

**Review:**

The paper studies the impact of content moderation policies in online communities.
The authors provide a framework to explain how a platform's choices for its content moderation policy affect the dynamics of the users and online communities.
Their proposed model provides a vocabulary and mathematically tractable framework for analyzing platform decisions about content moderation.



(+) The paper targets a very important problem.

(+) The authors provide a detailed survey of the related work.

(+) The paper is clearly written and easy to follow.


(-) The contributions of the paper are somewhat unclear.

(-) There is no experimental evaluation using real datasets.

(-) The relevance of the call for papers is slightly questionable.



Overall, the paper is clearly written and targets a very interesting problem.
The authors provide a detailed survey of the related work.
However, there are no references from the web conferences, and the paper does not show how to contribute to the web community.
Also, there is no experimental evaluation using real datasets.
It seems that there is a lack of empirical evidence to demonstrate the effectiveness of the proposed solution.
Additionally, I think the contributions of the paper are somewhat unclear.
It would be beneficial if the authors could include a contribution summary (or problem definitions) in the introduction section.

**Questions:**

N/A

**Reviewer Confidence:**

2: The reviewer is willing to defend the evaluation, but it is likely that the reviewer did not understand parts of the paper

**Scope:**

3: The work is somewhat relevant to the Web and to the track, and is of narrow interest to a sub-community

---

### Decision · Program_Chairs · 2024-01-22

**Decision:**

Accept (Oral)

**Comment:**

Summary: The paper introduces a theoreticla model and studies the impact of content moderation policies in online communities.

 Strengths:
 + well-motivated problem
 + interesting counter-intuitive results backed by theoretical proofs
 + real-world explanatory examples

 Weaknesses:
 - no empirical evaluation using real datasets
 - some concern about Web relevance
 - some concern about length of appendix as final paper has limit

 Recommendation: Well-executed paper. Practical applicability and Web relevance can be strengthened.